# When Two-Fold Is Not Enough: Quantifying Uncertainty in Low-Copy qPCR

**DOI:** 10.3390/ijms26167796

**Published:** 2025-08-12

**Authors:** Stephen A. Bustin, Sara Kirvell, Tania Nolan, Reinhold Mueller, Gregory L. Shipley

**Affiliations:** 1Medical Technology Research Centre, Faculty of Health, Education, Medicine and Social Care, Anglia Ruskin University Chelmsford, Chelmsford CM1 1SQ, UK; 2The GeneTeam, Bury St Edmunds IP31 1AA, UK; 3RM Consulting, San Diego, CA 92131, USA; 4Shipley Consulting, Vancouver, WA 98682, USA

**Keywords:** qPCR, relative quantification, low target concentrations, amplification efficiency, measurement uncertainty, MIQE

## Abstract

Accurate interpretation of qPCR data continues to present significant challenges, particularly at low target concentrations where technical variability, stochastic amplification, and efficiency fluctuations confound quantification. The widespread assumption that qPCR outputs are intrinsically reliable, coupled with inconsistent adherence to best-practice guidelines, has exacerbated issues of reproducibility and contributed to misleading conclusions. This may distort pathogen load quantification in diagnostic settings, whilst in gene expression studies, it can lead to overinterpretation of small fold changes. This study presents a systematic, cross-platform evaluation of qPCR performance across a wide dynamic range using defined reaction mixes and technical replicates. We show that calculated copy numbers can closely match expected values over more than three orders of magnitude, but that variability increases markedly at low input concentrations, often exceeding the magnitude of biologically meaningful differences. We conclude that establishing and reporting confidence intervals from the data itself is essential for transparency and for distinguishing reliable quantification from technical noise.

## 1. Introduction

Although quantitative PCR (qPCR) [1,2] is a widely used reference method for detecting and quantifying nucleic acids [3,4], concerns about data reliability persist yet are generally overlooked [5]. One key reason lies in qPCR’s very accessibility. Its apparent simplicity, commercial availability, and ubiquity have paradoxically become liabilities [6]. In many studies, qPCR is treated as a secondary method to validate high-throughput data, without the statistical and methodological rigour expected of primary analytical platforms. Yet qPCR is not a “quick confirmation” tool; it is a quantitative measurement system that demands the same analytical scrutiny applied to microarrays or next-generation sequencing [7]. Treating it as an afterthought undermines the science it is intended to support and contributes to widespread underappreciation of measurement uncertainty, particularly at low target concentrations [5,8,9,10].

Measurement uncertainty arises from multiple sources, including fluctuations in amplification efficiency, background fluorescence noise, and technical variability introduced through operator handling, pipetting imprecision, and differences in reagent and instrument performance [11,12,13,14,15,16,17,18]. These effects are further magnified at low target concentrations, where stochastic amplification bias introduces additional, unpredictable variability [19], making it difficult to distinguish genuine biological signal from technical noise when interpreting small fold changes in gene expression or pathogen load [11,20].

Three commonly overlooked factors contribute to this problem. First, underreporting of variability: few studies report even standard deviations (SD) or coefficients of variation (CV) for fold changes, let alone 95% confidence intervals (CI), despite their importance for assessing biological relevance. Second, arbitrary replicate design: most qPCR experiments default to three technical replicates without statistical justification, even though high-Cq (>30 cycles) targets may require five or more replicates to account for Poisson noise [21]. Third, validation bias: when used to confirm high-throughput results, qPCR experiments often omit efficiency corrections, increasing the risk of false-positive findings [11].

Inconsistency within molecular diagnostics can result in misleading assessments of pathogen load [22,23], with potential consequences for clinical decision-making [24]. In gene expression studies, inadequate validation may lead to technical variability being mistaken for genuine biological effects [19], particularly when interpreting small changes [25]. Unlike microarrays and RNA sequencing, which incorporate statistical corrections for technical noise [26,27], qPCR lacks a systematic approach to modelling or correcting for measurement uncertainty. Whilst there are guidelines aimed at promoting transparency and methodological consistency in qPCR experiments [28], adherence remains sporadic, especially regarding the reporting of measurement uncertainty [8,19,29,30]. As a result, arbitrary thresholds for significance are often applied, even though these may fail to reliably distinguish biological variation from technical noise [12,30].

This study presents a series of systematically controlled experiments demonstrating how platform-specific variability, input concentration, and efficiency assumptions affect quantification reliability. Our analysis focuses on variability in copy number calculations and fold change estimation and how this influences data interpretation, particularly at low target concentrations. Through empirical and model-based analyses, we identify where qPCR precision deteriorates and demonstrate that CIs should be empirically derived from each dataset to distinguish true signal from technical noise. This approach enables more cautious and transparent interpretation of small fold changes and strengthens the reproducibility of qPCR experiments in both diagnostic and research settings.

## 2. Results

### 2.1. Technical Characterisation of Assays

Absolute copy numbers for standard curve generation were determined via droplet digital PCR (ddPCR), ensuring accurate baseline quantification (Figure 1). qPCR efficiency and standard curve linearity were assessed on the BioRad Opus with serial dilutions of ddPCR-quantified amplicons for the seven assays. All assays demonstrated high linearity (R^2^ ≥ 0.99) and amplification efficiencies ranging from 92% to 99%, confirming optimal qPCR performance.

LoD values were determined by testing 24 technical replicates at 50, 20, and 5 copies per reaction (Appendix A). Limit-of-detection (LoD) studies confirmed reliable detection at 20 copies per reaction for five assays and at 50 copies for the remaining two (Appendix A).

### 2.2. Pipetting Accuracy and Low Volume Amplification

Pipetting accuracy was assessed using a single reaction mix containing approximately 3 × 10^3^ copies/µL of *GAPDH*-A, which was amplified in 1, 2.5, 5, 10, and 20 µL reaction volumes on the Bio-Rad (Hercules, CA, USA) CFX Opus, with twelve replicates per condition in a 96-well plate. This setup was not intended to mimic standard workflows but to isolate pipetting variability by holding all other reaction components constant. Using a single reaction mix allowed us to assess how transfer volume alone affects amplification consistency. The results demonstrated that small volumes could be dispensed with sufficient precision and that Cq values were highly consistent across 2.5–20 µL volumes (Figure 2A). In contrast, the 1 µL reactions exhibited markedly increased variability, with multiple non-detections and high dispersion among the remaining replicates, indicating poor quantitative reliability at this volume.

To evaluate whether this pattern persisted across platforms and at lower template concentrations, the experiment was repeated using the same reaction mix on three different block-based instruments. The findings confirmed that reliable quantification was maintained from 2.5 µL upward, whereas 1 µL volumes again yielded inconsistent and often undetectable results (Appendix A). An additional titration with 1–12 µL volumes in 1 µL increments further demonstrated that amplification efficiency and Cq reproducibility remained stable across this extended volume range, reinforcing the conclusion that small-volume reactions (≥2 µL) are feasible when handled carefully (Appendix A).

We prepared four sets of reaction mixes. Set A contained approximately 2 × 10^2^ copies/5 µL of *GAPDH*-A, and Set B contained approximately 8 × 10^2^ copies/5 µL of *ACTA2*-A. These were run in twelve 5 µL replicates on the Bio-Rad Opus and BMS Mic (Figure 2B,D,E). Sets C and D contained the same concentrations but were run in 20 µL volumes (Figure 2C,F,G). Results were consistent across reaction volumes and instruments (Figure 2H), with ∆Cq values between 5 µL and 20 µL reactions remaining small and reproducible.

### 2.3. Reproducibility of Replicate Assays

We prepared a set of four independent reaction mixes, each containing approximately 1 × 10^7^ copies/5 µL of *ACTA2*-A (Set A). Three additional sets were prepared at lower concentrations: Set B (2 × 10^5^), Set C (5 × 10^3^), and Set D (5 × 10^2^ copies/5 µL). Each set included four independently prepared reaction mixes, each run in 24 technical replicates on the Bio-Rad Opus (n = 96 per set). Amplification curves were tightly grouped with minimal variation, and mean Cq values (±SD) remained consistent across replicate groups (Appendix A).

To assess the influence of assay design on reproducibility, we prepared three additional sets targeting *CDKN1A* and *IGF-1* in duplex reactions, using diluted breast cancer cDNA. These were detected using Texas Red (*CDKN1A*) and Cy5 (IGF-1) probes. Each concentration set included seven technical replicates, ran on the Opus (Appendix A), and repeated on the Mic using five replicates (Appendix A); the smaller number of replicates was due to it having a 48-well block. The *CDKN1A* assay showed low variability across conditions (SD 0.07–0.21), while *IGF-1* amplification was more variable (SD 0.20–0.55) (Appendix A). This pattern persisted across both platforms. One outlier reaction in the *IGF-1* duplex was excluded from analysis. Despite greater variability at low template input, Cq values remained within a controlled range, supporting the feasibility of quantification with careful assay design.

### 2.4. Inter-Instrument Uniformity

A comparison of inter-instrument uniformity between three block-based thermal cyclers showed that Cq values for *GAPDH*-A and *ACTA2*-A coamplified in 96 wells were tightly clustered on each instrument (Appendix A). This comparison is not intended to reflect a biologically meaningful relationship between *GAPDH* and *ACTA2* but to illustrate how technical variability alone can affect ΔCq and inferred fold changes. The point is not to interpret *GAPDH* vs. *ACTA2* per se, but to highlight that such a difference can arise purely from inter-platform variability, reinforcing our broader argument that technical noise can confound inference in qPCR studies unless rigorously accounted for. Intra-instrument variability in ΔCq values ranged from 1.4 to 1.7 (Appendix A), with a pooled ΔCq of 1.5 (Appendix A). This corresponds to a 2.9-fold expression difference, exceeding the commonly used two-fold threshold for biological significance. These results confirm that intra-instrument reproducibility is high, supporting consistency within a given platform. However, the inter-instrument differences, while modest, indicate that platform-specific effects alone can produce what would be biologically meaningful shifts in ΔCq. This highlights the importance of rigorous standardisation in inter-platform qPCR studies.

### 2.5. Quantification Accuracy

To characterise reaction performance across platforms, we prepared a set of reaction mixes designated Set A (high input), Set B (moderate input), and Set C (low input). Each set comprised four samples. This experiment is designed to assess whether small, defined fold differences can be reliably distinguished under conditions that vary both input amount and instrument platform. In Set A, Sample 1 contained 4 × 10^5^ copies/5 µL of *C. auris* gDNA, as quantified by ddPCR. Samples 2, 3, and 4 contained 2×, 3×, and 4× that concentration, respectively. This fold difference series was generated by substituting water in the qPCR premix with proportionally increased amounts of input DNA. In Set B, Sample 1 contained 4 × 10^3^ copies/5 µL, and in Set C, 4 × 10^2^ copies/5 µL. As in Set A, Samples 2, 3, and 4 in Sets B and C contained 2×, 3×, and 4× the concentration of Sample 1, respectively. All samples were amplified on the Bio-Rad Opus using twelve technical replicates per condition, generating 4 × 12 = 48 Cq values per set. Data were analysed using the instrument’s default threshold settings (Figure 3A–C). Within each set, amplification curves were tightly grouped and demonstrated clear stepwise separation. Calculated copy numbers exhibited low variability, with CVs ranging from 4% to 13%, with greater variability observed at the lowest template input.

Three additional sets were generated using the same structure and amplified on the Mic instrument. In Set D, sample 1 contained 8 × 10^3^ copies/5 µL, and in Sets E and F, 5 × 10^2^ copies/5 µL (Figure 3D–F), all quantified by ddPCR. Again, samples 2, 3, and 4 contained 2×, 3×, and 4× the concentration of sample 1. Amplification kinetics closely resembled those observed on the Opus platform, with overall CVs slightly lower across the full concentration range.

Quantification accuracy was assessed by comparing measured copy numbers to expected values across all individual replicates. Pearson correlation analysis is summarised in Figure 3G. On the Opus platform, all three sets (A–C) showed perfect correlation (r = 1.0, R^2^ = 1.0), with narrow CIs and highly significant *p* values. On the Mic instrument, correlation coefficients for Sets D–F ranged from 0.93 to 0.99 (R^2^ = 0.87–0.99). All were statistically significant (*p* < 0.05), except for Set F (r = 0.93, *p* = 0.066), where low template input introduced greater variability. However, combining data from Sets E and F (n = 8) restored statistical significance (r = 0.95, *p* = 0.0003), indicating that quantification remained reliable at a low copy number despite modest run-to-run variation. These results directly address our central aim: to determine whether small, known fold differences can be accurately resolved across a range of input levels and instrument platforms. They demonstrate accurate and reproducible quantification across more than three orders of magnitude, with consistent performance across both platforms and minimal technical variability at moderate to high template inputs.

Quantification accuracy was further visualised by plotting log_2_-transformed expected versus measured copy numbers for each input level (Appendix A). Each point represents the mean copy number derived from 12 technical replicates, with error bars indicating 95% CIs. Results showed tight agreement with expected values across more than three orders of magnitude, with increasing variability at low input concentrations, particularly on the Mic platform.

To assess quantification resolution at low target concentrations (Cq > 30), we compared qPCR performance across three instrument platforms using three independently prepared sets of reaction mixes. Each set comprised four samples, with Sample 1 containing the lowest *C. auris* gDNA concentration (50–100 copies per reaction). Samples 2, 3, and 4 contained 1×, 2×, and 3× more target, respectively. Amplifications were performed on the Bio-Rad CFX Connect (Figure 4A) and Opus (Figure 4B) using 16 replicates per sample and on the BMS Mic (Figure 4C) using 12 replicates, again limited because of the 48-well block. Copy numbers were calculated using a standard curve generated on the Opus platform and applied uniformly across all instruments.

All platforms reliably detected low target levels and returned narrow 95% CIs. However, the Connect exhibited a ΔCq offset of approximately 1.5 cycles relative to the Opus for Sample 1, which resolved at higher input concentrations. This offset distorted fold difference estimates and led the Connect to overestimate copy number differences (Figure 4D). While none of the platforms precisely quantified the expected doubling of the target in Sample 2, the Opus correctly captured the 3x increase in Sample 4. The Mic instrument also performed well, maintaining close agreement with expected values for Samples 3 and 4. These findings reveal subtle platform-specific differences in quantification at low template input and demonstrate the importance of instrument validation when interpreting small fold changes near the assay’s detection limit.

### 2.6. Accuracy of Relative Fold Differences

Gene expression analysis commonly reports the abundance of target genes relative to one or more reference genes as fold differences. To evaluate how instrument performance affects qPCR-derived fold change measurements, we prepared three reaction mixes using human fibroblast cDNA and assays targeting both *GAPDH*-A and *ACTA2*-A. Mix 1 contained undiluted cDNA; Mix 2 was a 1:15 dilution; and Mix 3 was a 1:500 dilution. Amplifications were performed on the Bio-Rad CFX Connect (Figure 5A) and BMS Mic (Figure 5B), with 16 technical replicates per dilution.

ΔCq values between diluted and undiluted samples closely matched the expected dilution factors on both platforms. Fold differences, calculated from regression-derived copy numbers, were internally consistent within each instrument and exhibited overlapping 95% CIs (Figure 5C). However, fold change estimates systematically differed between platforms. This discrepancy was attributable to small but consistent differences in ΔCq values recorded between the FAM and HEX detection channels. Although mean Cq values differed by only 0.27 cycles, the Mic reported fold differences approximately 1.65-fold higher than the Connect. This likely reflects platform-specific differences in fluorescence detection and Cq calculation algorithms. This comparison is not intended to imply any biological relationship between *GAPDH* and *ACTA2*. These targets are used here solely as convenient assay pairs to test how platform-dependent factors, such as Cq detection thresholds, dye-channel separation, and standard curve calibration, affect the reproducibility and interpretability of qPCR-derived fold differences. The entire study is focused on technical sources of variability; no biological inference is drawn from these results.

These results were obtained using duplex reactions, where the target and reference genes were amplified in the same well (intra-assay quantification). However, relative quantification is more commonly performed using singleplex reactions, where fold difference quantification occurs across separate wells (inter-assay quantification). To compare these approaches, we designed two experimental setups.

First, intra-assay variability was assessed using a single premix containing both *GAPDH* and *ACTA2* assays. This mix was divided into two portions, one receiving undiluted breast cancer cDNA (Figure 5D) and the other a 1:5 dilution (Figure 5E). Each condition was amplified in 48 technical replicates. Second, inter-assay variability was evaluated using two independently prepared sets of singleplex reactions. The first set (Figure 5F) comprised 12 samples with undiluted cDNA: six with *GAPDH*-A primers/probe and six with *ACTA2*-A. The second set (Figure 5G) followed the same format but used 1:5 diluted cDNA. Each reaction was performed in four replicates.

Both approaches yielded consistent amplification profiles, with similar mean Cq values and reproducible ΔCq values. Fold differences were calculated by reporting *GAPDH* copy numbers relative to those of *ACTA2*, either within the same well (duplex) or across singleplex reactions. As shown in Figure 5H, intra-assay (duplex) quantification yielded narrower 95% CIs and lower CVs, particularly at lower input levels, indicating improved measurement reproducibility.

These findings confirm that duplex assays can improve qPCR precision in fold difference analysis, especially for small expression changes. However, this benefit is not universal. An alternative duplex combination (*GAPDH*-B/*ACTA2*-B), tested at low target concentrations, demonstrated higher variability in *ACTA2*-B amplification (Appendix A). The resulting fold difference, expressed relative to *ACTA2*-B, exhibited a much wider 95% confidence interval and a coefficient of variation of 37% (Appendix A). These results emphasise the need to validate duplex assay performance individually, as even minor changes in primer/probe design can affect amplification kinetics and compromise quantification accuracy.

To evaluate the reproducibility of fold difference quantification using multiplex normalisation, we measured a defined four-fold change in *CDKN1A* expression while *GAPDH*-A, *ACTA2*-A, and *IGF-1* levels were held constant. The observed fold differences, normalised to the geometric mean of these three targets, were consistent across platforms and yielded narrow 95% confidence intervals (Appendix A).

### 2.7. Accuracy of Detecting Small Fold Differences in Gene Expression

To further assess the ability of qPCR to detect small fold differences in gene expression relative to a reference gene, two sets of reaction mixes (Sets A and B) were prepared, each comprising four samples. In both sets, *ACTA2* serves as a fixed internal reference across all samples, enabling a controlled evaluation of how reliably small fold differences in *GAPDH* copy number can be detected under varying template inputs. In Set A, Sample A1 contained *GAPDH*-A (15,000 copies/µL) and *ACTA2*-A (1000 copies/µL) amplicons. Samples A2, A3, and A4 contained two-, three-, and four-fold higher concentrations of *GAPDH*-A, respectively, with no *ACTA2*-A. Set B followed the same format but at lower concentrations: Sample B1 contained *GAPDH*-A (300 copies/µL) and *ACTA2*-A (100 copies/µL), and Samples B2, B3, and B4 contained two-, three-, and four-fold more *GAPDH*-A, respectively.

Reference samples (A1 and B1) were amplified as duplex reactions, while test samples (A2–A4 and B2–B4) were amplified as singleplex assays on the Bio-Rad CFX Duet (16 replicates/sample; Figure 6A,D) and the BMS Mic (12 replicates/sample; Figure 6B,E). Cq values were determined using each instrument’s default threshold settings. Copy numbers were calculated using regression equations derived from standard curves.

For intra-assay fold difference quantification, *GAPDH*-A copy numbers in A1 and B1 were expressed relative to *ACTA2*-A within the same duplex reaction. For inter-assay quantification, copy numbers in the remaining test samples (A2, A3, A4, and B2, B3, B4) were expressed relative to the average *ACTA2*-A copy number measured in the corresponding reference sample (A1 or B1). Fold differences were then calculated relative to Sample 1 for each set (Figure 6C,F).

In Set A (moderate copy number), the calculated fold differences closely matched the expected two-, three-, and four-fold increases. CIs were narrow and encompassed the expected values, indicating that qPCR can reliably resolve small fold changes when template input is sufficiently high. In Set B (low copy number), fold differences remained directionally consistent with expectations, but CIs were broader, reflecting reduced measurement precision. This loss of resolution is consistent with increased stochastic noise at low input, where small fluctuations in the abundance of one of the target genes disproportionately affect fold difference estimation. These findings demonstrate that accurate detection of small fold differences is feasible using qPCR, but reliability is reduced at low target abundance. Interpretation of small expression differences under such conditions requires cautious validation and replication.

### 2.8. Accuracy and Reliability of Fold Change Estimates Deteriorate with Increasing Cq and Inaccurate Efficiency Assumptions

All Cq values recorded in Figure 3, Figure 4, Figure 5 and Figure 6 served as the empirical basis for bootstrap simulations to estimate the uncertainty associated with fold change measurements at different template abundances. For each dilution pair (e.g., 1× vs. 2×), six Cq values per condition were randomly sampled with replacement from the original dataset. A ΔCq value was then calculated as the difference between the mean Cq values of the two sampled groups. For example, if the sampled means were 26.9 (1×) and 25.6 (2×), then ΔCq = 26.9 − 25.6 = 1.3. This value was then converted to a fold change using the amplification equation: Fold change = (1 + E)^ΔCq^, where E = 0.992 (corresponding to 99.2% efficiency). This sampling and calculation process was repeated 10,000 times per comparison, generating a distribution of fold change values. The number 10,000 was chosen to ensure stable estimation of confidence intervals while preserving computational efficiency. From each bootstrap distribution, 95% confidence intervals were derived to reflect the uncertainty of the estimated fold change at a given template abundance.

As shown in Figure 7A, fold change precision deteriorated markedly with increasing Cq. While intermediate Cq values (25–30) exhibited slightly narrower confidence intervals than the <25 group, this reflects greater amplification variability among the highest input (4×) samples within the <25 bin, rather than improved performance at lower input. At high target input (Cq < 25), empirical fold differences were tightly constrained, with CI widths of 2.2–4.0, closely matching theoretical expectations. Precision declined at intermediate Cq values (25–30), where CI widths ranged from 1.4 to 2.7. Above Cq 30, however, CIs widened substantially: for example, the 95% CI for the 1× vs. 3× comparison spanned from 0.71 to 14.74, despite a nominal three-fold difference in input. These Cq brackets were chosen to reflect commonly encountered categories in the qPCR literature, specifically abundant transcripts (e.g., reference genes), moderately expressed targets, and low-abundance or near-threshold detections. They are not intended as strict thresholds but are examples of practical groupings used to aid interpretability. Our empirical data highlight that at low template concentrations, stochastic sampling dominates technical noise, impairing the reliability of even large fold differences.

To independently assess how amplification efficiency affects fold change estimation, we modelled ΔCq values from 1.0 to 6.0 using three representative efficiencies: 92%, 100%, and 110%. These values were chosen to reflect commonly cited upper and lower limits of the acceptable efficiency range (Figure 7B). CIs were estimated analytically by propagating a ΔCq SD of 0.2 cycles, isolating the effect of efficiency alone. Fold change estimates remained similar at ΔCq ≤ 2 but diverged rapidly at higher values. At ΔCq = 4, the estimated fold change ranged from 11.3 (92% efficiency) to 23.0 (110%), compared to 16.0 at 100%. At ΔCq = 6, values ranged from 50.1 to 85.8, a 1.7-fold divergence driven entirely by efficiency assumptions.

These data demonstrate that fold change estimates become increasingly unreliable when high Cq values are accompanied by inaccurate amplification efficiencies. They highlight the importance of reporting amplification efficiency and CIs when interpreting relative expression data, particularly at low template concentrations or when small fold changes are considered biologically meaningful.

## 3. Discussion

qPCR’s reputation as a “basic” technique stems from four interrelated factors. First, its apparent technical simplicity, a “plug-and-play” illusion promoted by instrument manufacturers and commercial kit suppliers’ pre-mixed reagents, creates the impression that successful assays require little more than template addition and button-pushing. Second, the default output from most qPCR instruments presents amplification curves without efficiency diagnostics, offering limited insight into assay quality. Third, automated Cq assignment by proprietary software can create false confidence in data accuracy. Finally, widespread reliance on the 2^ΔΔCq^ method [31,32] conceals multiple potential sources of error behind a deceptively simple calculation.

Despite over three decades of widespread use, variability in qPCR data is not always fully recognised or transparently reported. The MIQE guidelines were introduced to address these concerns and promote consistent reporting. However, uptake varies, and important methodological details affecting data quality are still frequently absent from the published literature. In clinical settings, unquantified variation can lead to misclassification of pathogen load and inappropriate treatment decisions. In gene expression studies, small differences interpreted without adequate uncertainty estimates risk overstatement of biological effects. The MIQE guidelines were developed to address these challenges by promoting transparency, reproducibility, and methodological rigour in qPCR studies [28,33]. However, adherence remains inconsistent, and key parameters affecting data quality, particularly those influencing measurement uncertainty, are often omitted.

### 3.1. Understanding qPCR Variability: Sources and Consequences

Experimental variability in qPCR arises from multiple sources, including reagent composition, pipetting imprecision, operator inconsistencies, and instrument-specific factors. These effects are amplified at low target concentrations, where stochastic sampling and amplification efficiency fluctuations introduce substantial uncertainty [11,12]. By combining ddPCR-based absolute quantification with empirically derived efficiency values, we established robust standard curves for all seven qPCR assays (Figure 1), achieving high linearity (R^2^ ≥ 0.99) and amplification efficiencies within the acceptable range (92–99%). Limit-of-detection (LoD) studies confirmed reliable detection at 50 copies per reaction across all assays (Appendix A). Together, these benchmarks form the basis for evaluating assay performance and the interpretability of fold change data.

### 3.2. Technical Replicates and Statistical Power

There are two types of replicates in qPCR [34]: biological replicates, which capture sample-to-sample variability, and technical replicates, which control for procedural noise arising from pipetting, reagent handling, and instrument performance. The latter are the focus of this study; however we stress that for small fold changes, or low copy number targets, increasing the number of biological replicates is also advised. Although three technical replicates are commonly used by convention, this choice is rarely supported by statistical justification. The optimal number of replicates depends on the level of technical noise, particularly at high Cq values, where stochastic sampling dominates.

A more rigorous approach incorporates statistical power analysis based on expected effect size and observed Cq variability, coupled with error propagation that accounts for amplification efficiency uncertainty. For example, assuming a standard deviation of 0.5 Cq, detecting a 1.5-fold change (Δ = 0.58 Cq) with 95% confidence and 80% power would require six technical replicates. Replicate number should therefore be empirically optimised to ensure the observed fold changes are both statistically and biologically interpretable.

### 3.3. Precision and Accuracy

qPCR precision refers to the reproducibility of Cq values under consistent conditions, while accuracy reflects how closely estimated values match true target abundance [35]. Our data show that precision can be high, but accuracy is strongly influenced by assay design, reagent quality, and instrument performance, especially at low copy number. Most studies report fold change or mean Cq values without quantifying uncertainty, increasing the risk of overinterpreting marginal differences. This highlights the importance of using consistent, well-defined metrics to measure and interpret technical variation in qPCR results.

### 3.4. Reaction Volumes and Instrument Performance

Reaction volume and instrument performance were evaluated using both 5 µL and 20 µL reactions on block-based and rotary instruments. Consistent amplification and low inter-replicate variability were observed across platforms and volumes ≥2.5 µL, confirming pipetting accuracy and reliable quantification at reduced reaction volumes (Figure 2 and Appendix A). Although 1 µL reactions occasionally produced amplification, they were characterised by high failure rates and large dispersion, rendering them unsuitable for quantitative analysis.

When template concentration was doubled incrementally (e.g., 2 → 4, 3 → 6 copies per reaction), the corresponding ΔCq values were markedly compressed (0.06–0.25 cycles), far below the theoretical one-cycle shift expected with 100% efficiency. This compression reflects a combination of low target copy number, stochastic template distribution, and potential efficiency loss near the detection threshold. An alternative possibility is that Cq values may track target concentration more closely than absolute copy number when reaction components are held constant across volumes. This would suggest that relative fluorescence thresholds can partially decouple from molecule number under low-volume conditions, with amplification kinetics shaped by both stochastic effects and reaction geometry. While this remains speculative, it merits further investigation, as it may have implications for interpreting Cq shifts in small-volume assays. Importantly, since Cq values at each volume were internally consistent, these data confirm that small reaction volumes can be used with confidence in qPCR workflows, provided template input is sufficient and technical variability is controlled. While this setup does not reflect standard laboratory workflows, where individual cDNA samples are typically added to pre-prepared master mixes, our aim was to isolate and visualise the effects of low template input under highly controlled conditions. The observed dispersion at low copy number, even with uniform input and reaction setup, highlights the inherent stochasticity of qPCR at the lower limits of detection and underscores the need for appropriate replication and interpretation safeguards. Although the number of replicates used exceeds routine practice, it was chosen to support bootstrapped confidence intervals and to illustrate the full extent of technical variability, not as a recommendation for everyday assay design.

Reproducibility was further validated using independently prepared reaction mixes for *ACTA2* and duplex assays targeting *CDKN1A* and IGF-1. Despite increased variability at low input, Cq distributions remained controlled (Appendix A), reinforcing the importance of assay design in minimising technical noise.

Inter-instrument uniformity was evaluated across three Bio-Rad block-based systems. Although Cq values for co-amplified targets were tightly clustered within instruments, ΔCq variation between platforms approached the commonly accepted two-fold biological threshold (Appendix A), highlighting the potential for platform-specific bias in comparative studies. This effect is particularly important given the widespread assumption that equivalent assays yield comparable values across instruments, which these results challenge.

To assess quantification accuracy, six concentration series were amplified on two platforms using *C. auris* gDNA as a defined template. Calculated copy numbers closely matched expected values across more than three orders of magnitude (Figure 3), with high precision maintained down to a few hundred copies per reaction. This was corroborated by log_2_-transformed regression plots (Appendix A), where increasing variability at low input was reflected in wider confidence intervals. Platform-specific effects were further explored at very low target concentrations (Cq > 30). Although all instruments reliably detected the input, subtle offsets in ΔCq values, especially on the CFX Connect, led to distortions in relative copy number estimates (Figure 4). These findings illustrate how even modest platform-specific variation can affect quantification at the lower limits of detection. Direct comparison of Cq values across different instruments is therefore not valid without additional standardisation. Including a small set of calibrator samples on every plate offers a practical strategy for identifying and adjusting for plate- or instrument-specific shifts, and we recommend this approach as good practice. These results reinforce the need to report uncertainty metrics alongside fold difference values and to apply appropriate statistical frameworks when interpreting borderline qPCR data.

### 3.5. Same-Well Fold Difference Quantification: A Practical Strategy

We next examined the use of duplex qPCR to estimate fold differences in gene expression. Duplex reactions yielded lower variability and narrower confidence intervals than corresponding singleplex reactions (Figure 5), supporting same-well quantification as a practical strategy when precision is critical. However, this benefit proved assay-dependent: an alternative duplex combination (*GAPDH*-B/*ACTA2*-B) showed reduced reproducibility due to increased variability in *ACTA2*-B amplification (Appendix A), underscoring the need for assay-specific validation.

Although same-well duplexing may appear attractive for normalisation, caution is warranted: relying on a single reference gene is susceptible to both biological and technical variability. To evaluate whether accurate fold difference quantification could be achieved with proper normalisation, we prepared two defined samples differing only in *CDKN1A* concentration (4-fold higher in sample 2), while *GAPDH*-A, *ACTA2*-A, and *IGF-1* were held constant. *CDKN1A* was then normalised against the geometric mean of the three constant targets. This multiplex approach, applied on both the Bio-Rad Opus and BMS Mic platforms, yielded consistent fold change estimates with narrow 95% confidence intervals and low coefficients of variation (Alternative Appendix A). These findings support the utility of well-designed multiplex qPCR for precision applications, provided that reference targets are carefully validated and normalisation strategies are robust.

Finally, the ability of qPCR to resolve small fold differences in gene expression was tested using defined input ratios (2×, 3×, 4×). At moderate template concentrations, fold differences aligned closely with expected values, with narrow confidence intervals (Figure 6). At lower input, fold change estimates remained directionally correct but exhibited broader confidence intervals, consistent with increased stochastic variability.

### 3.6. Impact of Technical Variability on Accuracy

We used bootstrapping to estimate confidence intervals for fold change values because standard parametric methods do not adequately capture the structure of technical noise in qPCR experiments. Fold changes are nonlinearly derived from Cq values and are particularly sensitive to small errors when target abundance is low or ΔCq is small. Analytical error propagation typically assumes normality, homoscedasticity, and independence, assumptions frequently violated in practice. Bootstrapping avoids these assumptions by resampling observed data to empirically model the full distribution of possible outcomes. This distribution-free approach allows for the construction of confidence intervals that incorporate both raw Cq variability and the distortions introduced during their transformation into fold change. This is especially important where small differences may be interpreted as biologically meaningful but, in fact, lie within the bounds of technical uncertainty. Its flexibility and empirical nature make bootstrapping particularly suited to qPCR, where platform-specific variability, efficiency deviations, and non-normal error structures are common.

To quantify the impact of technical variability across the qPCR dynamic range, we applied a bootstrap resampling strategy using replicate Cq values for *GAPDH*-A and its empirically determined amplification efficiency (E = 99.2%). Fold change estimates were relatively precise below Cq 25, with narrow 95% confidence intervals and close agreement with theoretical fold differences (Figure 7A). In the mid-Cq range (25–30), precision remained high and, in some cases, exceeded that of lower Cq data, possibly reflecting optimal amplification kinetics or reduced batch-specific variability. However, above Cq 30, confidence intervals widened substantially, and fold change estimates became increasingly variable. For example, in a 1× vs. 3× comparison, the 95% CI ranged from 0.71 to 14.74, despite a mean estimate of 4.56.

We also modelled the impact of amplification efficiency on fold change estimation across a range of ΔCq values using three commonly cited efficiency values (92%, 100%, and 110%) to reflect the practical extremes of qPCR performance (Figure 7B). Confidence intervals were held constant to isolate efficiency effects. Fold change estimates aligned closely at ΔCq ≤ 2 but diverged progressively thereafter. At ΔCq = 4, fold change values ranged from 11.3 (92%) to 23.0 (110%), compared with 16.0 at 100% efficiency, a shift large enough to affect interpretation in experiments where 1.5- to 2-fold differences are considered meaningful. Even minor discrepancies, such as 3.8-fold versus 4.0-fold at ΔCq = 2, can alter conclusions when data lie near interpretive thresholds.

These results show that a considerable number of reported fold changes may fall within the range of bootstrapped uncertainty, making it difficult to distinguish them confidently from background variation. Moreover, even narrow confidence intervals may mask underlying imprecision in quantification when copy numbers are low. This distinction between precision and accuracy is especially critical when evaluating small expression changes. Our findings reinforce the importance of reporting statistical confidence alongside effect size in qPCR studies and caution against interpreting fold changes without explicit reference to their associated uncertainty. Table 1 summarises when amplification efficiency corrections are most critical.

### 3.7. Redefining Fold Change Thresholds: A Data-Guided Strategy

Statistical significance alone does not guarantee biological relevance. Small but precise fold changes may be inconsequential, whereas larger, noisier changes may reflect meaningful underlying biology. Our findings support replacing arbitrary fold change cut-offs with empirically derived thresholds based on actual assay performance. The following recommendations are grounded in the data reported in this study:Cq < 25: two-fold differences can be reliably detected.Cq 25–30: ≥three-fold differences are advisable.Cq > 30: ≥four-fold differences should be required unless high measurement precision is demonstrated.

These tiered thresholds reflect technical limits under controlled conditions. However, we emphasise that they are context-specific and ideally should be determined empirically for each assay system. Researchers should apply similar data-driven analyses to define appropriate thresholds within their own experimental systems, rather than relying on historical convention.

Amplification efficiency also plays a critical role in determining whether fold changes are meaningful or misleading. Even small deviations from 100% efficiency can introduce biologically relevant error, depending on the magnitude of the Cq difference (ΔCq) between samples. Three distinct regimes can be identified:Low ΔCq (≤1.5): Systematic bias dominates. Minor efficiency misestimation (e.g., assuming 100% when true efficiency is 92%) can shift a 2-fold difference to 1.7- or 2.3-fold, sufficient to alter biological interpretation when evaluating marginal effects.High ΔCq (≥3): Error propagation dominates. Efficiency deviations are amplified during logarithmic transformation, leading to substantial fold change underestimation and wider confidence intervals (as demonstrated in Figure 7B). Empirical measurement of efficiency is essential in this regime to avoid misleading conclusions.Intermediate ΔCq (1.5–3): Neither bias nor propagation fully dominates, but the risk of interpretive error remains. Here, empirical efficiency assessment still improves accuracy, especially for low-copy targets or marginal expression differences.

These observations are summarised in Table 1, which provides a decision framework for when efficiency correction is most critical. In all regimes, estimating efficiency improves quantitative interpretability, particularly when small differences are reported or large fold changes are inferred from wide Cq separations.

Data visualisation also contributes to interpretability. Bar charts, even when annotated with error bars, often obscure important variability by summarising data into a single central tendency. This is particularly problematic when interpreting marginal fold differences, where data spread, outliers, or distribution shape may materially affect biological interpretation. Alternatives such as scatter, box, or violin plots reveal the distribution of replicate measurements and help communicate the extent of technical variability. This more transparent presentation of replicate-level data, alongside confidence intervals, strengthens the evidential basis for any quantitative conclusion.

### 3.8. Sources of Technical Variability and Their Mitigation Across the qPCR Workflow

Upstream variability, from sample selection to reverse transcription, remains a major contributor to total technical variation [36,37]. Whilst assay optimisation and data analysis are the main focus of attention, pre-analytical steps often receive less scrutiny. However, these early stages account for a significant proportion of total technical variability in quantitative molecular workflows, particularly when RNA quality is variable [38,39,40], inhibitors are present [41,42], or RT enzyme performance is suboptimal [43,44]. Critically, this upstream noise is not correctable by downstream qPCR optimisation: even the most precise amplification system cannot recover information lost due to poor RNA integrity or inefficient reverse transcription [45]. This is especially true for low-abundance transcripts, where stochastic sampling effects further exacerbate variability [45,46]. To support improved assay reliability, Table 2 summarises the main sources of technical variability across the entire qPCR workflow, from pre-analytical handling through to amplification and analysis, outlines practical recommendations for mitigation, and serves as a general framework for improving reproducibility across applications.

These findings reinforce core principles: that high Cq values indicate greater uncertainty; that fold change estimation depends critically on accurate efficiency assumptions; and that low-level expression changes require more rigorous validation and transparent reporting.

### 3.9. Study Limitations

Although this study offers a comprehensive evaluation of qPCR performance, some caveats must be noted:

Sample types and biological variability: Synthetic and cell-derived templates enabled controlled assessment of technical variability. However, real-world samples introduce additional variability, particularly during RNA extraction and reverse transcription, which can affect assay accuracy in clinical and environmental settings. Future studies should extend these findings to a broader range of biological materials.

Inter-laboratory validation: Although results were consistent across platforms, validation was limited to a small number of instruments and controlled conditions. Collaborative studies across laboratories adhering to the MIQE guidelines will be needed to establish standard benchmarks for comparing instruments and technical reproducibility.

Context-specific fold change thresholds: The proposed precision-guided thresholds are context-dependent. A ≥4-fold difference cutoff is supported for low-abundance targets but may be impractical in high-throughput workflows and could be relaxed where precision is demonstrably high. Standardising how precision is assessed, potentially through instrument-specific benchmarks or harmonised protocols, remains a priority.

Regulatory implications: Precision-based interpretation challenges current regulatory frameworks that rely on fixed thresholds. Integrating uncertainty-aware metrics into clinical validation will require alignment with agencies such as the FDA and EMA.

Assumptions in bootstrap modelling: While our bootstrap approach enhances uncertainty estimation, it relies on simplifying assumptions, including fixed efficiency and normally distributed error. Future work should incorporate real-world variation and test the approach in more biologically complex contexts.

## 4. Materials and Methods

### 4.1. Reagents and Instruments

Details of all commercial reagents, plasticware, and instruments used in this study are provided in Appendix A. qPCR reactions were performed on 96-well block-based instruments (Bio-Rad Opus, Bio-Rad, Hercules, CA, USA; Duet, Bio-Rad, Hercules, CA, USA; and Connect, Bio-Rad, Hercules, CA, USA) and a 48-tube rotary qPCR platform (BMS Mic, Bio Molecular Systems, Upper Coomera, Australia). Digital PCR (dPCR) experiments were conducted using the Bio-Rad QX200 droplet digital PCR (ddPCR) system (Bio-Rad, Hercules, CA, USA).

### 4.2. Primers and Probes

All assays were designed using Beacon Designer v8.21 (Premier Biosoft, San Francisco, CA, USA). Target sequences for human mRNAs and *Candida auris* small subunit ribosomal RNA were retrieved from the NCBI database (https://www.ncbi.nlm.nih.gov/). Primer and probe specificity, and potential off-target amplification, were evaluated in silico using Primer-BLAST (https://www.ncbi.nlm.nih.gov/tools/primer-blast/, accessed on 7 May 2024) and BLAST (https://blast.ncbi.nlm.nih.gov/Blast.cgi, accessed on 7 May 2024). Oligonucleotides were synthesised and lyophilised by Sigma-Aldrich, reconstituted in sterile RNase-free water to 100 µM, and stored at −20 °C. All are listed in Table 3.

### 4.3. Nucleic Acid Targets–cDNA

Total RNA was extracted from human fibroblast or breast cancer cell lines using Qiagen RNeasy kits (Qiagen, Hilden, Germany). RNA integrity was confirmed using an Agilent Bioanalyser 2100 (RIN > 9) (Agilent, Santa Clara, CA, USA). The absence of inhibitors was verified using a ten-fold dilution series assayed by RT-qPCR of *GAPDH*-A and *ACTA2*-A using a Clara Probe one-step master mix (PCR Biosystems) (PCR Biosystems, London, UK). cDNAs were synthesised in 20 µL reactions from 250 ng total RNA using UltraScript reverse transcriptase with random hexamer priming (PCR Biosystems) (PCR Biosystems, London, UK). Reverse transcription was performed at 42 °C for 15 min, followed by inactivation at 85 °C for 10 min. The resulting cDNA was diluted to 50 µL with RNase-free water and stored at −20 °C.

### 4.4. PCR Amplicons

Amplicons spanning the full target sequences were generated by 40-cycle PCR reactions using the respective forward and reverse primers, without DNA-binding dyes or hydrolysis probes. Products were serially diluted to a 1:10^6^ working stock using two successive 1 µL-to-1 mL transfers in RNase-free water. The purpose of this serial dilution was to generate a final dilution suitable for accurate ddPCR quantification, not to determine copy numbers at intermediate steps. To bring concentrations within the quantifiable range of ddPCR, further dilutions of the 1:10^6^ stock were made by pipetting 10 µL into 90 µL (1:10 or 1:100, as needed). ddPCR was then used to quantify this final dilution, and the resulting copy number was used to assign a value to the preceding 1:10 dilution, which served as the highest concentration standard for the qPCR curve. All subsequent qPCR standards were prepared by serial 1:10 dilutions using 10 µL into 90 µL transfers. All dilutions were performed in an adjacent laboratory, in a biosafety cabinet, to minimise contamination.

### 4.5. Droplet Digital PCR (ddPCR)

PCR amplicons and *C. auris* gDNA were quantified by ddPCR using Bio-Rad ddPCR Supermix (Bio-Rad, Hercules, CA, USA). Reactions (22 µL total volume) included primers and probes at final concentrations of 0.9 µM and 0.25 µM, respectively. Due to spectral detection limitations on the QX200 platform, *CDKN1AA* and *IGF-1* were detected using FAM- and HEX-labelled probes with the same sequences as those listed in Table 3. Droplets were generated using a DG8 cartridge and the QX200 droplet generator (Bio-Rad, Hercules, CA, USA). Twenty microlitres of reaction mix were dispensed into the middle wells, with 70 µL of droplet generation oil in the lower wells. Following droplet generation and sealing with pierceable foil, 40 µL of droplets were transferred into a 96-well ddPCR plate and amplified using a Bio-Rad C1000 thermal cycler (Bio-Rad, Hercules, CA, USA). Cycling conditions were: 95 °C for 10 min; 40 cycles of 95 °C for 15 s; and 60 °C for 60 s, followed by an enzyme deactivation step at 98 °C for 10 min. Droplets were analysed using the QX200 droplet reader (Bio-Rad, Hercules, CA, USA).

### 4.6. qPCR

qPCR reactions were prepared using RapiDxFire Probe Mix (Biosearch Technologies, Hoddesdon, UK), with final concentrations of 0.5 µM primers and 0.2 µM probes. Three experimental designs were implemented to assess precision: instrument uniformity (same reaction mix pipetted into 96 wells), intra-assay variability (multiple replicates from a single mix), and inter-assay variability (independent mixes per replicate group, as defined per experiment). These designs optimised statistical power while conserving reagents and minimising the influence of outliers. Although the number of replicates in many experiments exceeds what would be practical in routine workflows, these designs were selected to support downstream statistical analyses (e.g., bootstrapping) and to illustrate the extent of technical variability under controlled conditions. Unless otherwise stated (e.g., experiments shown in Appendix A), all qPCR reactions were performed in 5 µL volumes. Cycling involved an initial 2 min denaturation, followed by 40 cycles of 95 °C (denaturation) and 60 °C (extension). Bio-Rad instruments used 1 s steps for both phases, whereas the Mic platform used 5 s denaturation and 10 s extension steps. Reactions were conducted in heat-sealed 96-well plates (block-based instruments) or clear four-tube strips (Mic instrument).

### 4.7. Quantification of DNA Amplicons

Serial dilutions of PCR-generated amplicons were used to establish standard curves by qPCR, with absolute copy numbers for each dilution determined by ddPCR. Cq values were plotted against log_10_-transformed ddPCR-derived copy numbers. PCR efficiency (%E) was calculated from the slope of the standard curve using the following formula: %E = (10^(−1/slope)^ − 1) × 100. Copy numbers for unknown samples were calculated using the linear regression equation: 10^((Cq − y-intercept)/slope)^.

### 4.8. Bootstrap Simulation of Fold Change Variability

To estimate the precision of fold change measurements across the dynamic range of qPCR, empirical bootstrap simulations were performed using replicate Cq values obtained from experiments presented in Figure 3, Figure 4, Figure 5 and Figure 6. Three representative dilution sets were selected to span low, medium, and high Cq ranges (<25, 25–30, >30). Each set comprised 12–16 technical replicates at four target concentrations (1×, 2×, 3×, and 4×). To estimate variability in fold change comparisons, we used a bootstrap approach. For each pairwise comparison (e.g., 2× vs. 1×), we randomly sampled six Cq values with replacement from each of the two concentrations being compared. A ΔCq value was then calculated from each pair of sampled means. This procedure was repeated 1000 times per comparison, resulting in a distribution of ΔCq values representing the uncertainty associated with each fold change estimate. Fold change was computed from each ΔCq using the exponential amplification model: Fold change = (1+E)^∆Cq.^, where E is the empirically determined amplification efficiency (0.992, or 99.2%). The 95% CIs were calculated as the 2.5th and 97.5th percentiles of the bootstrapped fold change distribution. All resampling and calculations were performed in Python (v3.10) using NumPy and SciPy libraries. This empirical approach preserves the variability structure of the original experimental data and quantifies the impact of technical noise on fold change precision at different input concentrations.

### 4.9. Modelled Impact of Amplification Efficiency on Fold Change Estimation

To evaluate how amplification efficiency influences fold change estimation as ΔCq increases, theoretical modelling was performed using arbitrarily defined ΔCq values ranging from 1.0 to 6.0 in 0.1 cycle increments. Fold change was calculated using the same exponential model as described above. Three amplification efficiencies were modelled: 95%, 99.2%, and 110%. Confidence intervals were estimated analytically by propagating a fixed standard deviation of 0.2 cycles in ΔCq, assuming normally distributed error. This approach generated upper and lower bounds for each fold change estimate and allowed isolated evaluation of efficiency effects independent of empirical variability. To facilitate direct visual and quantitative comparison with the bootstrap-derived empirical results, the modelled confidence intervals were uniformly scaled to a median relative width of approximately ±27%, matching the typical uncertainty observed in the bootstrapped data, thereby enabling meaningful assessment of trends in fold change distortion without confounding from differing uncertainty scales.

### 4.10. Data Analysis

Unless otherwise specified, qPCR data were analysed using the instruments’ native software with default threshold settings. Raw data were exported to Microsoft Excel for Mac (v16.89.1 or 16.91) and analysed using GraphPad Prism for Mac (v10.3.1 or 10.4). ddPCR data were processed using QuantaSoft v1.6.6.320 and QX Manager v2.1 (Bio-Rad, Hercules, CA, USA).

## 5. Conclusions

An appropriately optimised qPCR assay is accurate, precise, and robust, delivering reproducible quantification across a broad range of template concentrations. However, even under controlled conditions, technical variability can confound the interpretation of small fold differences unless measurement uncertainty is explicitly quantified. Our findings support a data-guided strategy that incorporates empirically determined efficiencies, replicate-aware analysis, and confidence interval reporting to enhance interpretive confidence. Whilst further work is needed to validate this approach in complex biological environments, it offers a practical advance in qPCR interpretation that is fully aligned with the transparency and reproducibility goals of MIQE 2.0.

## Figures and Tables

**Figure 1 ijms-26-07796-f001:**
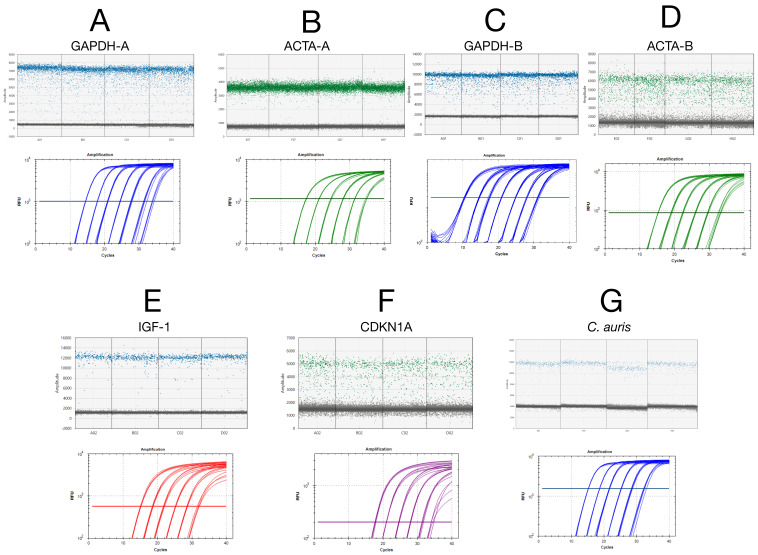
Absolute quantification of PCR amplicons and standard curves generated for the seven qPCR assays used in this study. ddPCR quantification was used to calibrate absolute input for qPCR. Each panel shows a ddPCR plot of fluorescence amplitude vs. event, with the standard curve below. Four ddPCR replicates were included to confirm pipetting consistency and ensure that downstream qPCR variability could not be attributed to pipetting error. (**A**) *GAPDH*-A E = 99.2% (y = −3.341x + 38.65). (**B**) *ACTA2*-A E = 93.6% (y = −3.487x + 39.039). (**C**) *GAPDH*-B E = 95.2% (y = −3.443x + 38.299). (**D**) *ACTA2*-B E = 95.2% (y = −3.444x + 38.30). (**E**) *IGF-1* E = 98.8% (y = −3.35x + 37.84). (**F**) *CDKN1AA* E = 98.4% (y = −3.360x + 38.76). (**G**) *Candida auris* E = 96.2% (y = −3.416x + 38.807). Due to the QX200 system’s two-channel detection (FAM and HEX) limitation, *CDKN1A* and *IGF-1* probes were resynthesised with FAM and HEX reporter dyes, respectively, for standard curve purposes. All Cq values are listed in the Appendix A.

**Figure 2 ijms-26-07796-f002:**
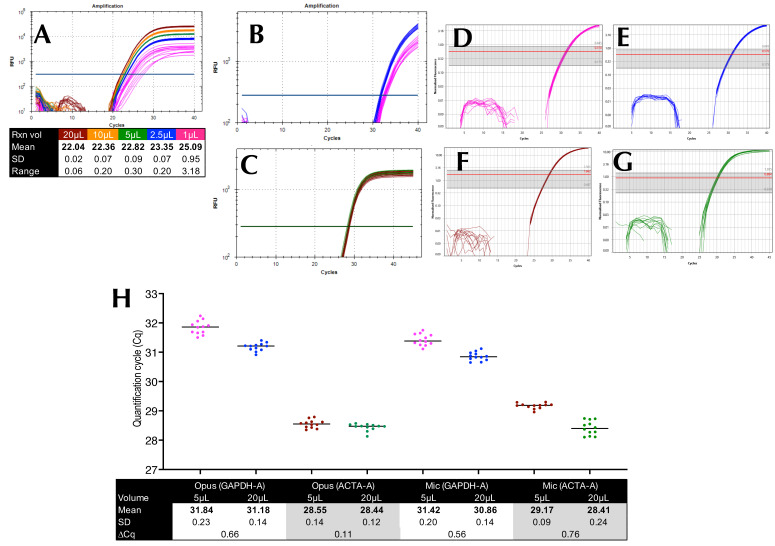
Pipetting accuracy for different reaction volumes. All reactions were carried out using twelve replicates. (**A**) Amplification plots and descriptive statistics for a run carried out with cDNA and targeting *GAPDH*-A using 1, 2.5, 5, 10, and 20 µL of the same reaction mix. (**B**) Amplification plots recorded for *GAPDH*-A with 5 µL (pink) and 20 µL (blue) volumes on the Opus. (**C**) Amplification plots recorded for *ACTA2*-A with 5 µL (brown) and 20 µL (green) volumes on the Opus. (**D**,**E**) Amplification plots recorded for *GAPDH*-A with 5 µL (pink) and 20 µL (blue) volumes on the Mic. (**F**,**G**) Amplification plots recorded for *ACTA2*-A with 5 µL (brown) and 20 µL (green) volumes on the Mic. (**H**) Plot of Cq values and descriptive statistics for all the reactions. All Cq values are listed in the Appendix A.

**Figure 3 ijms-26-07796-f003:**
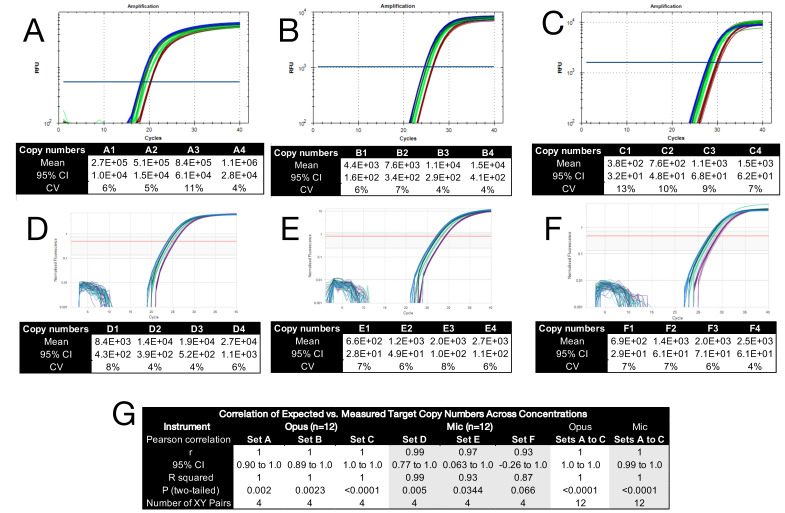
Amplification kinetics, quantification performance, and correlation with expected copy numbers across six reaction sets. Each set included four samples with increasing input concentrations, distinguished by the different colours. (**A**) Amplification plots and calculated copy numbers for Set A amplified on the Bio-Rad Opus. (**B**) Amplification plots and calculated copy numbers for Set B amplified on the Bio-Rad Opus. (**C**) Amplification plots and calculated copy numbers for Set C amplified on the Bio-Rad Opus. (**D**) Amplification plot and calculated copy numbers for Set D amplified on the BMS Mic. (**E**) Amplification plot and calculated copy numbers for Set E amplified on the BMS Mic. (**F**) Amplification plot and calculated copy numbers for Set F amplified on the BMS Mic. Amplification traces were generated using each instrument’s default threshold settings. Tables below each panel report the mean calculated copy number, 95% CI, and CV across 12 technical replicates per sample. CVs ranged from 4% to 13%, with increased variability at lower input concentrations. (**G**) Pearson correlation coefficients (r), R^2^ values, and two-tailed *p*-values comparing expected and measured copy numbers for each reaction set. All correlations were statistically significant (*p* < 0.05) except for Set (F) (Mic), where low input concentration reduced power. Combined analysis of Sets (E and F) yielded a significant correlation (r = 0.95, *p* = 0.0003), confirming reliable quantification at low copy number. All Cq values are listed in the Appendix A.

**Figure 4 ijms-26-07796-f004:**
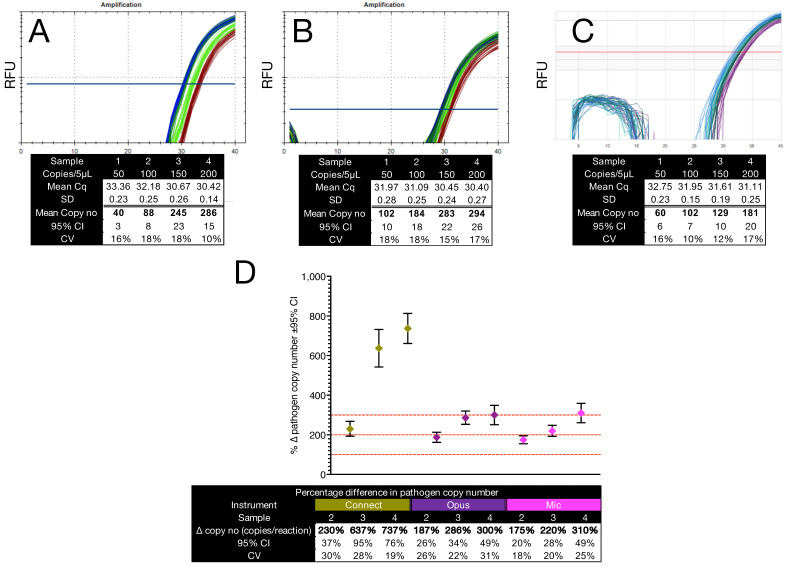
Accuracy of quantification at very low (Cq > 30) *C. auris* gDNA concentrations. Different colours denote the different copy numbers. (**A**–**C**) Amplification plots and descriptive statistics for quantification on the Bio-Rad CFX Connect (**A**), Bio-Rad Opus (**B**), and BMS Mic (**C**). Each panel shows amplification traces and tables summarising mean Cq, standard deviation (SD), estimated copy number, 95% CI, and CV, based on 16 replicates (Connect, Opus) or 12 replicates (Mic). Copy numbers were calculated using a standard curve derived from PCR amplicons on the Opus. (**D**) Percentage difference in calculated copy number for Samples 2–4 relative to Sample 1. Red dashed lines indicate expected time increases. Coloured symbols represent data from Connect (olive), Opus (purple), and Mic (pink). All Cq values are provided in the Appendix A.

**Figure 5 ijms-26-07796-f005:**
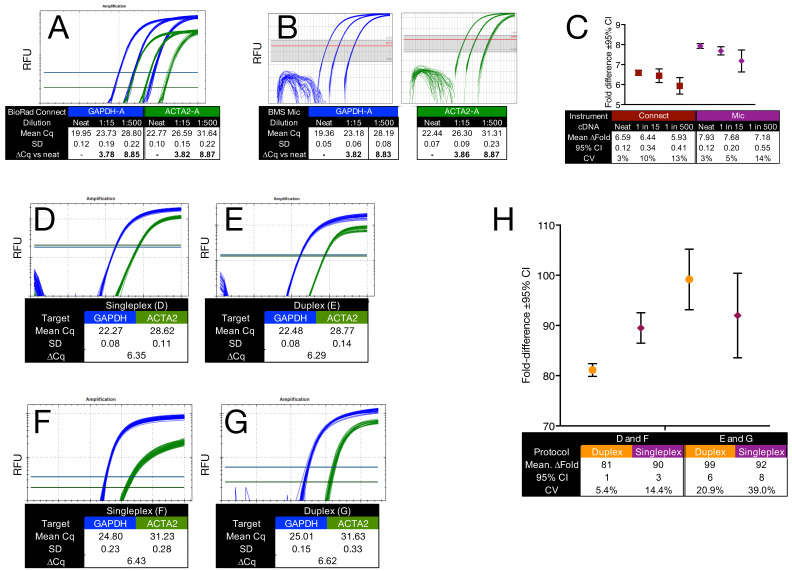
Instrument- and singleplex/duplex-dependent variability in qPCR fold difference quantification. (**A**) Amplification plots and descriptive statistics for duplex assays targeting *GAPDH* (blue) and *ACTA2* (green) obtained on the Bio-Rad CFX Connect. (**B**) Amplification plots and descriptive statistics for the same assays recorded on the BMS Mic. The Mic software does not permit simultaneous display of FAM and HEX channels. (**C**) Fold differences (±95% CI) for *GAPDH*/*ACTA2* at three tested dilutions and run on the BioRad Connect (brown) and BMS Mic (purple). Whiskers indicate 95% CIs. (**D**) Amplification plots and descriptive statistics for duplex reactions (*GAPDH*/*ACTA2*) at high/medium starting template concentrations. (**E**) Singleplex reactions at high/medium starting template concentrations, with *GAPDH* and *ACTA2* reactions prepared independently. (**F**) Duplex reactions at lower template concentrations. (**G**) Singleplex reactions at low template concentrations. (**H**) Calculated fold differences (±95% CI) for intra- and inter-assay comparisons. Duplex values are shown in orange, singleplex values in purple. Fold differences were calculated by reporting *GAPDH* copy numbers relative to those of *ACTA2*, as calculated from their respective standard curves. All quantification cycle (Cq) values are provided in the Appendix A.

**Figure 6 ijms-26-07796-f006:**
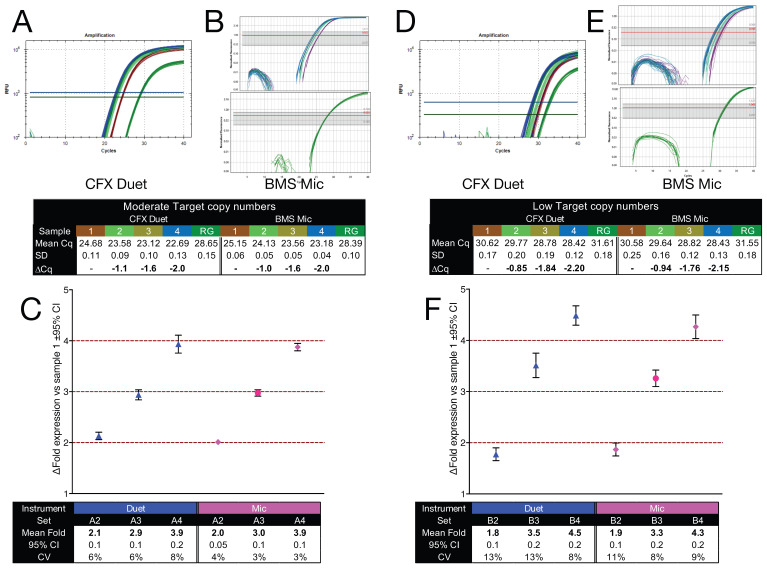
Detection of small fold differences in *GAPDH* expression relative to *ACTA2* at moderate and low target copy numbers. The different colours denote the different samples. (**A**,**B**) Amplification plots and descriptive statistics for Set (**A**) (moderate copy number), run on the CFX Duet (**A**) and BMS Mic (**B**). *ACTA2* is shown in green; *GAPDH* in brown, light green, olive, and blue to denote increasing concentration. (**C**) Calculated fold differences (±95% CI) for Set (**A**). Hatched lines indicate expected 1-, 2-, 3-, and 4-fold increases. (**D**,**E**) Amplification plots for Set (**B**) (low copy number), run on the Duet (**D**) and Mic (**E**,**F**). Calculated fold differences (±95% CI) for Set (**B**), with the same colour-coding and CI representation as in panel (**C**). All quantification cycle (Cq) values are provided in the Appendix A.

**Figure 7 ijms-26-07796-f007:**
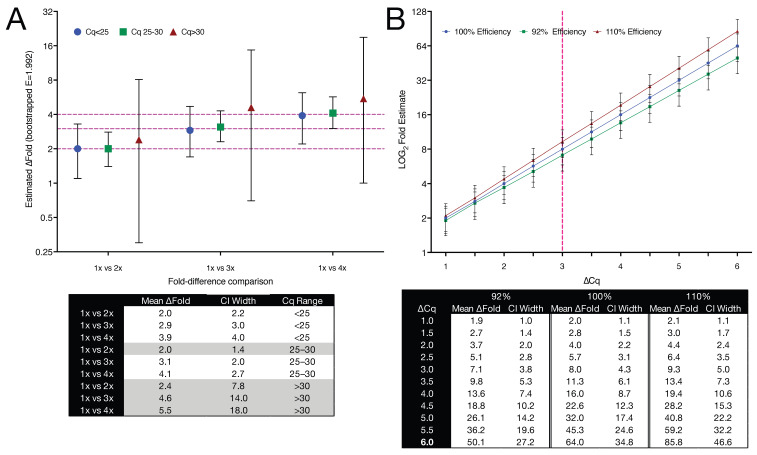
Impact of Cq range and amplification efficiency on fold change estimation and interpretability. (**A**) Empirical bootstrap-derived fold change estimates across three Cq ranges, based on 12 replicates at four input concentrations (1×, 2×, 3×, 4×). Data are shown on a log_2_ scale and stratified by Cq: <25 (blue circles), 25–30 (green squares), and >30 (red triangles). Dashed horizontal lines indicate theoretical 2-, 3-, and 4-fold differences. CI widths (95%) for each comparison are shown in the table below; shaded rows highlight comparisons where fold change uncertainty exceeds interpretability thresholds based on Cq bin. (**B**) Modelled influence of amplification efficiency assumptions on fold change estimates across ΔCq values from 1.0 to 6.0, using efficiencies of 92% (green), 100% (blue), and 110% (red). The y-axis is log_2_-transformed for interpretative clarity. CI widths were calculated assuming a fixed ΔCq SD of 0.2 cycles. The dashed vertical line at ΔCq = 3 denotes the point where even small efficiency deviations begin to introduce substantial interpretive bias. Tabulated values below indicate the mean fold change and corresponding 95% CI width for each condition.

**Table 1 ijms-26-07796-t001:** When does amplification efficiency matter?

Although 100% amplification efficiency may be a common assumption in qPCR workflows, this introduces systematic bias, particularly in contexts requiring accurate quantification of small fold-changes or low-abundance targets. The extent to which efficiency matters depends on three interrelated factors:			
** Factor **	** Effect on ∆Fold Accuracy **	** Explanation **	** Implication for Efficiency Handling **
ΔCq magnitude	↑ Greater ΔCq amplifies error	Efficiency errors scale exponentially with ΔCq. At ΔCq = 6, a 95% efficiency underestimates fold-change by ~17%.	Measure efficiency if ΔCq ≥ 3. Assuming 100% may be acceptable for ΔCq ≤ 1–2.
Efficiency deviation	↑ Larger deviation increases systematic bias	Even small deviations cause consistent under- or overestimation across all targets and replicates.	Empirical calibration is preferred unless assay conditions are known to support consistent efficiency.
Fold-change size	↑ Small changes are most affected	A 5% bias is negligible when comparing 10-fold changes but critical when distinguishing 1.5× from 2.0×.	Always measure efficiency when interpreting small fold-changes (<2×), especially with clinical or mechanistic implications.
Summary recommendation: Use empirically determined, target-specific efficiencies for any experiment where fold-change accuracy matters. Precision without accuracy is misleading.			

**Table 2 ijms-26-07796-t002:** Key sources of variability in qPCR and recommended mitigation strategies. Summary of experimental and analytical factors contributing to variability in qPCR results, their impact on quantification accuracy, and practical steps for mitigation. This table provides a reference framework for improving reproducibility in both gene expression and pathogen quantification applications.

Step	Source of Variability	Impact on qPCR Data	Recommendations
** 1 **	Pre-Assay Sample Handling	RNA integrity and extraction efficiency influence RT efficiency and overall assay precision	Standardise RNA extraction conditions across experiments; determine RNA integrity and quality; optimise RT conditions
** 2 **	Pipetting Precision	Is crucial to maximise precision	Use calibrated and appropriate volume pipettes; assess Cq variance across replicates
** 3 **	Reaction Volume	Affects sensitivity, fluorescence detection, and reagent efficiency	Validate lower volumes; use consistent volumes; consider automated pipetting
** 4 **	Instrument Performance	Inter-run variability can introduce measurement inconsistencies	Regularly assess instrument reproducibility; account for platform-dependent effects where possible.
** 5 **	Fluorescence Detection	Spectral overlap and detection sensitivity affect quantification accuracy	Verify fluorophore performance and quantification accuracy on the selected instrument
** 6 **	Primer/Probe Design	Suboptimal design can lead to inefficient amplification and non-specific signals	Optimise primers and probes for high specificity and efficiency
** 7 **	Amplification Efficiency	Variability in efficiency impacts accurate fold-change calculation	Amplification efficiency must be determined; correct for variations in amplification to ensure accurate quantification.
** 8 **	Normalisation Strategy	Across-well normalisation introduces more variability than same-well approaches	Prefer same-well normalisation (ideally multiplex assays) to reduce variability
** 9 **	Technical Replicates	Insufficient replicates increase uncertainty and lower confidence in results	≥3 technical replicates; for low-copy-number targets or when high precision is required, use 95% confidence intervals to indicate sufficient reproducibility

**Table 3 ijms-26-07796-t003:** Target genes, primers, probes, and amplicons used in this study. Associated amplicon quantification, PCR efficiency, and limit of detection (LOD) data are presented in Figure 1 and Appendix A. Uppercase letters in probe sequences indicate locked nucleic acid (LNA) bases. The designations A-F, A-R, A-Pr, and B-F, B-R, B-Pr refer to primer and probe sets A and B specific for *GAPDH* and *ACTA2*, respectively.

Target	Gene Symbol	NCBI Reference	Primers & Probes	Sequence (5′-3′)	Fluorophore	Amplicon (bp)
Glyceraldehyde-3-Phosphate Dehydrogenase	*GAPDH*	NM_002046.7	A-F	CGACAGTCAGCCGCATCTTCTTTTG	FAM	70
			A-R	TCACCTTCCCCATGGTGTCTGAG		
			A-Pr	cgcCagCcgAgcCaca		
			B-F	AGCCACATCGCTCAGACA	FAM	75
			B-R	TGACCAGGCGCCCAATAC		
			B-Pr	actCcgAccTtcAccttcc		
Alpha Smooth muscle actin-2	*ACTA2*	NM_001141945.3	A-F	CTATGCCTCTGGACGCACAAC	HEX	64
			A-R	GACATTGTGGGTGACACCATCTC		
			A-Pr	agaGtcCagCacGatgcc		
			B-F	CCTCTGGACGCACAACTGGCATC	HEX	75
			B-R	AGCCCTCATAGATGGGGACATTGTG		
			B-Pr	tgaCacCatCtcCagAgtcc		
Cyclin dependent Kinase	*CDKN1A*	NM_000389	F	CTTGGTTAATATGCACATACTCCA	Texas Red	98
			R	TGGGGACACTGAAGCAAAC		
			Pr	atgCtgGtcTgcCgcc		
Insulin-like growth factor 1	*IGF-1*	NM_001111283.3	F	CCCAAGACCCAGAAGTATCA	Cy5	87
			R	CTTGCGTTCTTCAAATGTACTTC		
			Pr	acaAgaAcaCgaAgtctca		
* C. auris *	rRNA	MN796100.1	F	ACGGATCTCTTGGTTC	FAM	50
			R	GTATCGCATTTCGCTG		
			Pr	cgcAtcGatGaaGaac		

## Data Availability

The original contributions presented in this study are included in the article and Appendix A. Further inquiries can be directed to the corresponding author.

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
