# Peer review of "When Two-Fold Is Not Enough: Quantifying Uncertainty in Low-Copy qPCR"

_ijms, 2025, doi:10.3390/ijms26167796_

Round 1
Reviewer 1 Report
Comments and Suggestions for Authors
The manuscript by Bustin et al is comprehensive, and addresses some important considerations/limitations of qPCR methods that are not necessarily widely appreciated. The lead author is one of the developers of the MIQE guidelines, and is thus widely known and well-regarded in qPCR circles: this manuscript should have been a straightforward “thumbs-up, good to go”.
Unfortunately, there are several problems with the authors’ conclusions, there are far too many (not necessarily helpful or informative) figures, and moreover the authors do not (to my mind) take sufficient care to explain their reasoning, leaving the entire manuscript feeling like something of a missed opportunity. Whether they choose to take my suggestions on this latter point or not, the manuscript is going to need some work. Apologies, in advance, for the following.
In essence, the authors are very good at qPCR, and have written a manuscript _aimed_ at people who are less good at qPCR, with the intention of helping those people get better at qPCR, and design more rigorous experiments which are reported more thoroughly. So far so good. The problem is that they have _pitched_ the manuscript at people who are also very good at qPCR: following the authors’ decisions, experiments and reasoning requires a strong grasp of the fundamentals, such that the actual experiments then become largely superfluous, or even baffling (“nobody has ever done qPCR like this, so why even test this?”).
For example, one question the authors investigate is (essentially) “Do Cq values vary depending on which PCR machine you use?”, to which the answer is yes. But…it was obviously going to be yes. Anyone who knows how qPCR works would expect the answer to be yes, and empirically confirming this doesn’t really tell us anything we didn’t already know. Meanwhile, people who would actually benefit from this information (i.e. those who are less good at qPCR) are unlikely to actually grasp the implications of this experiment because it is…not dumbed down enough. Much of this information isn’t actually _new_, but falls into that odd category of “unknown to novices, but considered self-evident by experts”: pitching it to those novices is a more helpful approach, therefore.
The manuscript would be more useful if it were restructured as (if you excuse the turn of phrase) “qPCR for dummies”: explaining the reasoning behind each experiment in simple terms, and concluding each section with a “lesson learned/suggested advice” segment. Basically “is X a problem? Yes, X is a problem. How can we avoid X? By doing Y.”
The authors have the clout to get away with presenting what are, essentially, well-known limitations, if those limitations are couched more simply, with helpful, idiot’s-guide tips and tricks alongside. I have provided examples of this throughout (suggestions only: the authors are free to workshop something better).
Working through the manuscript:
Introduction:
This is basically fine. The authors could, perhaps, stress the advantages qPCR has over RNAseq, in that yes, it can be used as a “quick confirmation” (though all the correctly-noted caveats apply to this) but it can also be used on a much, much larger panel of samples, very affordably. If RNAseq using N=3 suggests gene X is modestly upregulated, confirming this for N=200 is prohibitive via RNAseq, but pretty trivial via qPCR. qPCR can (and should) be used with large N values -more on this later.
2.1: technical characterisation
The idea of calibrating qPCR to ddPCR is good, since ddPCR can absolutely give absolute copy number values. Here, though, the authors miss an opportunity to directly compare the two approaches more broadly.
Firstly, the amplicons are generated by running a qPCR to completion, then diluting the result via two successive 1:1000 dilutions, followed further by 1:10 and 1:100 dilutions as necessary to get samples on the dynamic range ddPCR can handle. Given the very next section addresses that “1ul pipetting vols are highly inaccurate”, it seems odd to use 1ul vols not only once, but _twice_, to prepare standards. The authors are essentially just diluting speculatively in the assumption that ddPCR will sort it out anyway, which is…fine, I guess, but not a great look for a paper that specifically addresses issues like pipetting error.
(I usually gel extract to ensure single amplicon, then spec to obtain [DNA], and work out copies.ul-1 based on mass of my amplicon: this gets surprisingly close)
The authors also only measure ONE concentration of sample, but do so four times for each. In my experience, for the sort of copy numbers shown in figure 1, ddPCR is incredibly consistent: even duplicate ddPCR runs are largely pointless.
What would work very well here instead would be to demonstrate (using the same volumes and dilutions and pipetting steps) that ddPCR and qPCR agree very well over certain ranges, but that while ddPCR is much better at giving absolute numbers (that are not subject to efficiency differences, since ddPCR is an end-point assay), the dynamic range is much more limited.
So for GAPDH-A, the authors could present a full 10-fold dilution curve from (for example) a 1:1000 dilution of the neat PCR product all the way down to <1 template molecule per well, conducted via qPCR and ddPCR: this would show readers that above certain target concentrations, ddPCR is useless (all droplets are positive) while qPCR remains potentially viable. Meanwhile, below certain concentrations (stochastic partitioning range), both qPCR and ddPCR become more variable. The region spanned by the effective dynamic range of the ddPCR assay could then be used to determine the efficiency of the qPCR assay, and its upper and lower limits.
Here the “idiot-guide” lines would be something like “Accuracy and dynamic range differences between ddPCR and qPCR: when to use, and when not to use, each method, and how to use one to calibrate the other.”
2.2: pipetting.
This section is good, in that pipetting error is a major source of variation in qPCR, but the chosen approach is really odd. A single reaction mix, but pipetted at different volumes? Is that something anyone has ever done, as part of an experimental pipeline? I cannot envisage a scenario where this would be even tangentially sensible. It’s not even a practical way to run qPCR, since preparing distinct reaction mixes for each sample prior to dispensing just wastes time and masses of extra tubes. Most people make a mastermix of enzyme, buffer and primers (in one tube), dispense that to all the relevant wells, and then add cDNA individually to each well after that. Consequently, a “master mix + variable volumes of cDNA” would be a better demonstration of pipetting accuracy (and would likely reach the same conclusions: “what volume of diluted cDNA is appropriate to use? 1ul is highly error prone, but 2ul+ is fine.”
This section DOES, however, expose a really interesting result that the authors slightly allude to in the discussion, but which I feel deserves greater focus: what _would_ you expect if you used different volumes of the same mix? Naively, one might assume (as the authors do in the discussion) that you would see Cq differences commensurate with number of target molecules, and thus measuring the same reaction mix in 5ul, 10ul and 20ul vols would produce three distinct values, each 1 cycle earlier than the one before. On the other hand, these are all the same reaction mix, so the _concentration_ of molecules is identical, and since qPCR works on relative fluorescence change, perhaps the Cq values would be identical? As the results suggest, the latter appears closer to reality (also see figure S2), but not entirely: Cq values do decrease fractionally as _number_ of target molecules increases, but seem to still much more closely reflect _concentration_ of target molecules, which is really interesting. The authors say “this compression reflects a combination of low target copy number, stochastic template distribution, and potential efficiency loss near the detection threshold”, but…does it? The data from figure S2 shows that a ~6-fold difference in template (2ul-12ul) results in ~0.5 cycles of difference (should be ~2.6 cycles), which seems far more dramatic than can be attributed to stochastic partitioning or efficiency. There is something going on here that suggests both the hypotheses discussed above are not the whole picture (but that the latter might be closer).
Obviously the ‘for dummies’ take-home message here is “always use the same volume of reaction mix” (which frankly, shouldn’t need saying -though say it anyway), but it would be worth discussing these findings in more detail.
2.3: replicate assays
As noted above, most people do not prepare a massive amount of reaction mix + cDNA, and then pipette it out into 24 replicate wells, but instead individually add small amounts of cDNA to individual wells (perhaps 3 replicates per cDNA sample) containing premixed primers/enzyme/buffer. I am not sure, consequently, that this experiment really addresses any realistic practical scenario, other than showing that replicates are more variable at low copy number, which is certainly true. At reasonable copy numbers, replicate wells should be consistent (after all, if qPCR _wasn’t_ consistent between replicate wells, what the hell is the point of qPCR?)
The authors also use a ridiculous number of replicates (both here and throughout the manuscript). Replicate wells are essential for qPCR, certainly, but…12? 48? 96? I realise that much of their extensive replicate strategy is to produce data for their post-hoc bootstrapping analysis, but it would make more sense to declare this at the beginning, so readers don’t get the idea that 12 replicate wells are a good idea for day-to-day qPCR. It would, in fact, be an excellent opportunity to demonstrate, empirically, how accuracy changes with multiple replicates. For each dilution/target concentration, when do further replicates become superfluous?
“Should qPCR always be conducted in replicates? Yes! How many is appropriate for a given target concentration?”
2.4: inter instrument variability
This section absolutely is of benefit to “dummies”, but should be much, much blunter in explaining the rationale.
“Can you directly compare Cq values between different qPCR machines? No!”
This section is also written confusingly: inter-instrument and intra-instrument are both mentioned, but the focus appears to be exclusively on the former, making it unclear whether the latter is actually addressed at all, or is just a typo. I think examining intra-instrument variability is absolutely worth assessing too (and could be shown here). For this, again:
“Can you directly compare Cq values between two different plates on the same qPCR machine? No!”
All of these issues can, of course, be addressed by including a number of ‘calibration’ samples on every plate, and adjusting Cq values post-hoc, accordingly. This is something that should be more widely disseminated, and thus the authors should state this bluntly (rather than just alluding to “rigorous standardisation”).
“Can you nevertheless indirectly compare Cq values between two different plates or two different machines? Yes! By including a set of the same samples on every plate as calibrators (we recommend 4-5 samples): modest but plate-wide shifts in Cq values can be detected via calibrators and adjusted for accordingly”
2.5: Quantification accuracy
This section is highly confusing. Firstly, the reaction mixes prepared are stated as “2x, 3x or 4x the concentration”, i.e. the implication is the authors started with a small amount of template and then added more of it, somehow (or concentrated it)? When the focus is on how low copy numbers are less accurately quantified, presenting it this way gives the impression that the authors are “measuring something too rare to quantify accurately, then slowly increasing this inaccurate thing until it is measured accurately”, which raises all sorts of questions about the initial ‘inaccurate measurement’ the subsequent increases are predicated on. Especially since the authors say “sample 1 contained _approximately_ 4x10^5 copies”: hard to spin a convincing argument around “we started with this much, probably, and couldn’t quantify it accurately, either. But if we added more it got better, except the resulting numbers were all over the place because initial quantification values for the low amounts were inaccurate anyway”.
Consider instead presenting this as “lots of target, diluted until no longer accurately quantifiable”, since that more useful, biologically, and is fundamentally what the authors are trying to show. After all, anyone who starts by quantifying (poorly) a vanishingly small amount of target and then uses that value to calibrate all other (larger) values…really has no place in the lab, let alone doing qPCR.
Also, please reconsider the nomenclature for this section: Sample 1, 2, 3, 4 are all the same target in set A, but “set A” isn’t mentioned in the manuscript text, only the figure legend! Trying to work out what “B3” or whatever actually means is needlessly difficult.
Similarly, consistency: sometimes 2x/3x/4x is used, but sometimes “100%/200%/300% more” is used: these are the same numbers, but presented in confusing ways. Figure 4 doesn’t even reflect these changes accurately, either: it should be “50, 100, 150, 200 copies”, but the figure shows “50, 100, 150, 300 copies”, which suggests even the authors are confused by their terminology.
Take home message:
“Does accuracy decrease at lower target concentrations? Yes! Solution: use more replicates! (also see section 2.3). Also, always use HIGH abundance samples to calibrate LOW abundance samples, NOT the other way round.”
2.6: Fold-differences
This is another slightly weird one where the test conditions do not reflect any real laboratory scenario. Masses of reaction mix using a single cDNA sample, either concentrated, or diluted, then dispensed into a ludicrous number of replicates. When (and why) would anyone do this?
I also do not necessarily agree with the authors interpretations, either: directly comparing two distinct PCR reactions, whether duplex or singleplex, and expecting to derive a direct assessment of copy number difference…isn’t generally valid, and certainly shouldn’t be encouraged. A Cq of 23 for GAPDH and a Cq of 24 for ACTA2 doesn’t mean there’s twice as much GAPDH as ATCA2. There might be _approximately_ twice as much, maybe, but these are two different PCRs, and cannot be generalised. Numbers should be calibrated against a standard curve (generated using the same machine) and converted to actual copies before making this assessment. DeltaCq just isn’t appropriate here. Consider (as the authors do later) efficiency differences, which are reaction specific and which compound hugely over multiple cycles: take a reaction with efficiency of 95%, and one with 105%, and compare them to a ‘perfect’ efficiency of 100%. After 23 cycles at 95%, there are ~0.55x as many amplicons as at 100%. At 105%, ~1.76x as many. So, two different PCRs could produce the exact same Cq, but the underlying copy numbers could differ by 3-fold. Meanwhile, if two samples from the SAME PCR primer set give Cq values of 23, then regardless of efficiency, these samples contain very similar amounts of target.
If you add in duplex/singleplex distinctions on top of this, you’re adding “pipetting error” into the equation too, but as the authors show (and show in other experiments too) pipetting error is largely dismissible provided vols are 2ul+.
I am also not convinced that this data shows “duplex assays can improve precision, especially for small fold changes”, since the fold change being (questionably) assessed here is pretty substantial (~6-8 fold).
Really, the major take-home from this work is that you…probably shouldn’t do this sort of comparison (unless you’re very careful, know exactly what you’re doing, and make all the necessary post-hoc adjustments). For the audience most likely to benefit from this study, none of these necessarily apply, and this approach should probably not be encouraged.
2.7: Small fold-differences
This is very similar to section 2.5, and really, so is the odd sample preparation, and confusing and unhelpful nomenclature. The real take home here is “don’t use low abundance, hard-to-quantify-accurately samples to calibrate high abundance samples”, which should be largely self-evident. Sample preparation is questionable: the authors add to each reaction mix only what they are planning to detect, whereas for truly rigorous comparisons, both GAPDH and ACTA2 targets should be added whether the authors are detecting them or not (this can make a difference, even if simply through macromolecular crowding).
2.8: Accuracy/reliability with increasing Cq/efficiency
This section is ostensibly helpful, and represents the ‘core message’ of the manuscript, but is also…not particularly surprising, and nor are the conclusions necessarily correct. The authors run a massive bootstrapping analysis which is quite impressive, but ultimately conclude what should be fairly obvious: for low copy numbers, where more cycles are needed, efficiency contributions make a bigger difference (because differences compound exponentially). Also, for low copy numbers, where greater well-to-well variation would be expected through stochastic partitioning, variability between replicate wells is higher. Half of this is just an exercise in excel maths. And the conclusions are…not correct (discussed below).
The data moreover suggest that CIs increase with decreasing copy number, when they…don’t (they’re actually smaller for intermediate Cqs, and larger for low and high Cqs, pretty much across the board -which is interesting).
Discussion:
My biggest issue with the central thesis of this manuscript is that the authors are attempting to simultaneously argue that low copy numbers are hard to quantify accurately (I agree), that small fold-changes are harder to assess at low copy numbers (I agree), and that efficiency calcs are important for such comparisons (I disagree).
Efficiency problems only really become meaningful when the difference (between samples) in Cq values is large, but when the difference in Cq values is large…so is the underlying difference in expression. From a biological perspective, a 13-fold difference and a 20-fold difference are…not that different: both are ‘big fold changes’, and for something like a transcription factor (where large fold changes might occur) this range encompasses normal biological variation. Going from “some” to “lots” is biologically meaningful even if the precise quantification of the “lots” is slightly inaccurate. The authors claim that “Even apparently minor discrepancies, such as 3.8-fold versus 4.0-fold at ΔCq = 2, can alter conclusions when data lie near interpretive thresholds”, but I would…really disagree. A 4-fold change is pretty substantial, and inadvertently calculating this as a 3.8-fold change due to efficiency issues isn’t going to alter that fact. I personally would report a 3.8-fold change as “~4-fold”, anyway, because of the aforementioned biological variation. In Box 1 the authors say that “5% bias is negligible when comparing 10-fold differences, but critical when distinguishing 1.5 from 2.0x”, but at 2-fold difference, a 5% bias is…literally only 5%, since that’s one cycle.
Basically (in contrast to the suggestions in Box 1) when efficiency is a problem (high deltaCq), differences are too big for this to be of relevance. When differences are small, conversely, efficiency isn’t a problem (insufficient cycle differences between samples).
This seems so inherently obvious that I begin to wonder if I’m missing something. If so, I would be delighted to be corrected by the authors, but…I do not think I am.
Low copy numbers remain hard to quantify accurately, this I wholeheartedly agree, but I also disagree that for Cq >30, you should ‘need’ 4-fold differences. Instead, you should just…quantify those more accurately. The authors mention specifically how to do this (use more replicates) and they absolutely have the data to support this, given the sheer number of replicates they’ve used. You can measure low copy number targets if you measure them lots of times, because stochastic partitioning is correctable through simple averaging. You won’t get the same number of molecules in each well each time, so…measure more than three wells. Maybe six wells instead. Even ten if you want greater precision. If you’re quantifying targets that are so low that you often get negative wells (zero target), then…maybe just consider whether what you’re measuring is meaningfully present.
A much more useful message would therefore be “here is (approximately) how many replicates we recommend for each of these different Cq ranges, and this is the approx. point at which we would recommend you stop bothering entirely”.
All of which brings me to a major point that the authors just…don’t mention at all, meaningfully: biological replicates. Just…use big N values, for small changes. Or in general, really: for any study hoping to statistically assess data, bigger N values are better. With larger N values, it is also much, much easier to identify whether ‘small fold changes’ (say, less than 2-fold) are consistent, or are just biological noise. If a given gene does not differ between groups but does inherently vary by approx. 2-fold (just via transcriptional/biological noise) then this is much more obvious with N=10 than with N=3. If it _does_ differ between groups by ~2-fold, but also inherently varies by the same factor due to the aforementioned noise, then again this will be clearer at N=10. Measuring individual samples accurately is important, certainly, but accurate measurements of inherently noisy systems benefit hugely from high sampling frequency.
This also brings in another point the authors could make (which would be very helpful to the wider qPCR community, who do not, in my experience, necessarily make the correct interpretive leaps): placing results in biological context. Basically, understanding what the numbers mean.
The authors criticise 2^ddCT, and I agree completely here: taking raw, meaningful, informative data for control and test sample GOIs, and reference genes, and boiling all of that down to ‘fold change, probably’ is a terrible approach. For example, if sample A has GOI Cq 22 and ref Cq 24, while sample B has GOI Cq 29 and ref Cq 31, then the dCt values for A and B are both 2, and thus the fold change is zero. This completely misses the fact that _everything_ in sample B is ~7 cycles later than sample A, and probably something has gone very, very wrong with that sample.
Similarly, even without standard curves, Cq values reflect target abundance. A modest change (<2-fold) in Cq for something with a Cq average of 19 is likely to be real, since even that small change represents a huge number of transcripts. Meanwhile, a small (<2-fold) change in something with a Cq average of 29 is…probably not real, since that difference represents a mere handful of transcripts, and is much harder to measure accurately anyway -as demonstrated here (again, bigger N values would help assess innate biological variation and guide this).
All of this would be very helpful to discuss and summarise for the benefit of people who do not have a sufficiently strong grasp of qPCR fundamentals.
In summary, several of the core messages of this manuscript merit substantial revision/reconsideration, and indeed the entire text could be revised to render it more useful to audiences most likely to benefit from the messages within.
Further issues:
Figures: there are a lot of these (even more if you bring in the many supplementary figs), and virtually every figure includes a picture of the amplification curves. In many cases, this is ALL these figures show. These are not particularly helpful, especially when it’s essentially “48 replicates of the same reaction mix”, and thus are just one very fat green curve. Given the authors specifically note (lines 570-575) that scatterplots are more useful, it seems unfortunate that they have themselves chosen not to do this. The authors could render much of their data more immediately informative by simply presenting scatterplots of the Cq values, rather than pictures of the traces. Figure S2 does this well, for example.
Nomenclature: as noted above, the authors make some very odd choices in their chosen nomenclature, and how this is presented in figures: for example, figure 1: GAPDH-A is fig1A, ACTA2-A is fig1B, GAPDH-B is fig1C, and so on. For figure 6, there is “set A” being run on two different machines in 6A and 6B, while “set B” is presented in 6D and E. It is consequently somewhat difficult to work out which combination of which primers and dilutions each figure element refers to: “this is 6B, so obviously refers to set A, for GAPDH-A and ACTA2-A”. Even just adding the clarifying information to the figures themselves would be a great help.
Minor issues:
Figure 6 is present twice on my PDF copy. Formatting issue?
Table 1 should absolutely list “use consistent reaction volumes” as a tip, given the interesting results of that experiment. Consider adding “use calibration samples”, as well.
Author Response
Reviewer 1
We thank the reviewer for a detailed and perceptive critique that reflects considerable expertise in qPCR methodology. As noted, many of the limitations we address, such as instrument-specific variability, stochastic effects at low copy number, and the risks of interpreting small fold-changes, are familiar to experienced practitioners. Our aim was not to present these as novel observations, but to document them systematically and transparently, with a view to supporting didactic clarity and improving experimental practice.
We recognise the reviewer’s concern that some of the experimental setups may appear unnecessary or overly elaborate to those already proficient in qPCR. However, the fundamentals of experimental design and assay interpretation are not always applied consistently, even by researchers citing MIQE. The distinction, then, is not between ‘experts’ and ‘dummies’, but between theoretical familiarity and applied methodological rigour. By offering a detailed, replication-rich, and visually explicit analysis, we hope to bridge the gap between conceptual understanding and practical implementation, particularly for researchers less familiar with the quantitative constraints of low-copy qPCR. Since this paper is intended as a reference contribution to a special issue focused specifically on qPCR, it is designed to serve both novice and experienced users. Our aim is to provide a resource that clarifies the technical limitations of low-copy qPCR, illustrates how misleading results can arise even under seemingly well-controlled conditions, and supports critical evaluation of assay performance.
We have also reviewed the tone and presentation of the Discussion to ensure that it remains constructive and balanced. Our intention is not to criticise individual studies, but to provide empirical grounding for cautious interpretation and to support better-informed decisions about assay design and data analysis. We have therefore modified several passages to reinforce this didactic framing and to avoid language that might be read as overly negative or generalising.
We now respond to each of the reviewer’s points in turn.
- Introduction
The Introduction is essentially sound, and we acknowledge the reviewer’s point. Considerations of affordability and scalability are addressed later in the manuscript, particularly in relation to sample throughput and reagent use.
2.1: Technical characterisation
- There appears to be a misunderstanding of our dilution strategy. The serial dilutions were performed solely to generate a final sample within the quantifiable range of ddPCR; we did not attempt to determine copy numbers at intermediate steps. Once ddPCR provided an accurate copy number for this final dilution, we used it to assign a copy number to the preceding 1:10 dilution, which then served as the first point in the qPCR standard curve. This approach allowed us to anchor the standard curve to a reliably quantified reference. As clarified in the Methods, these final dilutions were performed using 10 µL into 90 µL, ensuring accurate volume transfers.
- We agree that ddPCR is highly consistent at the copy numbers shown and that duplicate or triplicate measurements are typically sufficient for quantification. However, our use of four replicates was intended to confirm the accuracy and consistency of our pipetting, given the broader focus of the manuscript on technical variability in low-copy workflows. We have clarified this in the legend to Figure 1.
- We appreciate the reviewer’s thoughtful proposal to compare qPCR and ddPCR performance across a broad dilution series. However, our aim in this manuscript was not to evaluate the relative merits or analytical characteristics of these platforms. Rather, we used ddPCR for a single, well-defined purpose: to obtain an accurate copy number for a reference dilution that would anchor our qPCR standard curves. Introducing a full cross-platform comparison would substantially expand the scope and length of the manuscript and introduce additional complexity not aligned with our core objective, which is to characterise variability within qPCR workflows at low copy number. We agree that such a comparison could be valuable in a dedicated methodological study, particularly to clarify where ddPCR’s quantification limits begin to impact interpretation relative to qPCR. However, given the focus of this study, we believe it is more appropriate to reserve such an analysis for future work.
- We believe there has been a misunderstanding of the intent and design of the experiment in question. The goal was not to emulate a standard qPCR pipeline, nor to replicate the typical distribution of master mix and cDNA across multiple tubes. Rather, our purpose was to isolate and quantify the variability introduced when pipetting different small volumes of the target template into a constant background, using a single master mix. By holding all other variables constant, including the composition of the master mix, the target concentration, and the amplification conditions, we aimed to evaluate the consequences of volume transfer alone on quantitative outcome.
This design allows for a clean assessment of pipetting error independent of other experimental steps, and highlights the disproportionate impact that small-volume inaccuracies can have on quantitative output, especially relevant when working at low template concentrations. In contrast, a more conventional “master mix + variable cDNA volume” design, whilst closer to typical lab practice, would conflate pipetting variability with other factors and therefore not permit the same controlled analysis.
We agree with the reviewer that in routine workflows it is standard practice to use a master mix and then add sample-specific cDNA volumes. However, our aim here was paedagogical rather than procedural: to make visible the extent of error introduced by pipetting 1 µL vs 2 µL vs 5 µL volumes, independent of other variables. This distinction is especially important in the context of low-copy qPCR, where small errors in input can lead to large fold-change artefacts.
We have clarified the rationale in the revised methods section to avoid confusion about the experimental purpose and to make explicit that this was not intended as a practical recommendation for reaction assembly, but as a focused investigation of pipetting-driven variability.
- The reviewer raises a valid question about what one might expect when using different volumes of the same reaction mix. This experiment was indeed designed to isolate the effect of pipetting volume using a single homogeneous reaction mixture. The results confirmed that Cq values primarily track target concentration rather than absolute copy number. Whilst we initially attributed the observed Cq compression to stochastic sampling and efficiency loss near the detection threshold, we agree that the modest shift across a ~6-fold change in input volume (Figure S2) deserves clearer discussion. We have revised the Discussion accordingly, acknowledging that concentration-dependent fluorescence accumulation appears to play a dominant role, and that the naïve expectation of a log-linear Cq shift based on input copy number does not hold under these conditions. We now explicitly state the practical implication: using a consistent input volume is essential to avoid introducing unintended variability into qPCR measurements.
2.3. Replicate assays
- We appreciate the reviewer’s further remarks concerning the design and relevance of our low-copy replicate experiment. We note that this comment differs from the earlier concern about pipetting accuracy, which focused on the use of varying reaction volumes to test volumetric precision. In that context, our intention was to isolate pipetting-related variability using a controlled, uniform reaction mix. By contrast, the present experiment involving 24 replicate wells seeded from a common reaction mix was designed not to reflect routine qPCR workflows, but to expose the extent of stochastic variation at low copy number under optimised and uniform conditions. This would of course not be done in real-life, but our objective was to illustrate that even under idealised circumstances, substantial variability can arise when template numbers approach the stochastic threshold. The exercise was not meant to simulate everyday laboratory protocols, but to provide a visual and quantitative reference point for appreciating the effects of low input on Cq dispersion.
We have clarified this distinction in the revised Discussion and reiterated that whilst qPCR is generally consistent at moderate-to-high copy numbers, interpretation becomes unreliable at low copy without sufficient replication or quantification safeguards. The practical implication that experimental design must anticipate and accommodate such variability remains a central message of the paper.
- The number of replicates used is indeed far in excess of what would be typical for day-to-day qPCR. This was intentional and necessary to generate a sufficiently large dataset for post hoc bootstrapping, particularly in low-copy contexts where technical variation is most pronounced. We now clarify this rationale at the start of the Methods to prevent any misinterpretation.
Whilst a formal analysis of the optimal number of replicates at different copy numbers is beyond the scope of this study, our data illustrate the diminishing returns of additional replicates at higher input levels. A brief note to this effect has been added to the Discussion, along with a clearer statement that our replication strategy was analytical, not prescriptive.
2.4: Inter instrument variability
We agree that direct comparison of Cq values across different instruments or even different plates on the same instrument is not valid without internal calibration. Our aim in this section was indeed to highlight that point, although we accept that the current wording may not make the rationale explicit enough. To clarify: this section addresses inter-instrument variability, not intra-instrument differences. We have removed the term “intra-instrument” to avoid confusion, as no within-instrument, between-run comparisons were performed in this part of the study.
We have now made the central message more explicit in the revised Discussion: Cq values are inherently platform-dependent, and even modest technical variation can result in misleading differences unless reference samples are included to calibrate across plates or instruments. We now clearly state that including a small number of shared calibrator samples (e.g., 4–5 per plate) enables post hoc correction for systematic shifts and allows meaningful comparison of Cq values between runs. This standardisation step is essential when experiments span multiple runs or instruments and is often overlooked.
2.5 Quantification accuracy
We acknowledge the reviewer’s concerns regarding the clarity of our description in Section 2.5 and agree that the framing of our experimental design may have been misleading. To clarify: we did not begin with a poorly quantified, low-abundance sample and attempt to improve accuracy by incrementally increasing template input. Rather, we first used ddPCR to establish the absolute copy number of a C. auris gDNA stock with high confidence. From this accurately quantified material, we then prepared a fold-difference series by substituting water in the qPCR premix with 2x, 3x, or 4x the amount of input DNA. The purpose of this experiment was to assess whether such small fold-differences could be resolved reliably by qPCR. All dilutions were therefore derived from a common, well-characterised stock.
We also accept the confusion caused by inconsistencies in naming and figure annotations. We have removed ambiguous labels (e.g., “Set A”) and replaced “Sample 1, 2, 3, 4” with clear numerical identifiers reflecting template input (e.g., 50, 100, 150, 200 copies). Inconsistent phrasing such as “2×/3×/4×” versus “100%/200%/300% more” has been standardised.
Finally, we fully agree with the reviewer’s practical summary: accuracy decreases at low input, and the solution is increased replication and appropriate calibration. Our experiment is explicitly designed to visualise this transition, and the key point , that high-abundance samples should be used to calibrate low-abundance ones , is now reinforced in the Discussion.
2.6: Fold-differences
- We hear the reviewer’s candid remarks and of course do not advocate that the described test conditions be repeated in routine laboratory workflows. The point of this experiment is precisely that it is not typical: it deliberately strips away extraneous sources of variability to expose the underlying technical noise in quantifying fold-differences. This study is not prescriptive—it is diagnostic. By using a single, uniform cDNA sample across a large number of replicates and defined fold-changes, we were able to highlight how precision erodes and how fold-change interpretation can become misleading, even under idealised conditions. That distortion is typically worse, not better, in standard practice. This rationale is now reiterated more forcefully in the Discussion.
- We appreciate the reviewer’s detailed analysis but wish to clarify a misinterpretation. We did not infer copy number differences by directly comparing the Cq values of unrelated assays. As stated in the manuscript, fold-differences were calculated by comparing GAPDH and ACTA2 copy numbers within duplex reactions or across matched singleplex reactions—following conversion to copy number. These values were derived using standard curves generated on the same instrument, as required for accurate quantification. At no point did we use raw Cq values from unrelated assays to estimate fold-differences. We agree entirely that efficiency differences can distort ΔCq-based comparisons, which is why our approach relies on calibrated, same-platform measurements. We have repeated that clarification in the legend the relevant phrasing in the text to prevent further confusion
- We respectfully disagree with the reviewer’s conclusion that these comparisons should not be encouraged. The purpose of this section was explicitly didactic: to quantify how fold-difference estimates behave under controlled conditions, including duplex versus singleplex formats. It is precisely because such comparisons are routinely attempted, often without sufficient understanding, that we believe they warrant empirical scrutiny.
Whilst the fold-differences tested here (∼6–8-fold) are not minimal, they are well within the range of differences commonly reported in gene expression studies. The fact that duplexing reduced technical variation, even at this level, reinforces the broader point that internal referencing within the same well when technically feasible offers a useful strategy to improve precision. As the reviewer notes, pipetting error was minimal across the experiments, and any minor variation was equally distributed between duplex and singleplex conditions. This reinforces our argument, rather than undermining it.
Finally, we fully agree that any attempt to infer fold differences between unrelated assays must be approached cautiously and, ideally, with calibration. However, that is not what was done here. All comparisons were made within a rigorously standardised setup using quantification against known input. The study does not encourage uncritical extrapolation but provides a data-driven framework for interpreting fold-change precision in low-noise systems. It needs to be read in parallel with the new MIQE 2.0 guidelines.
2.7: Small fold-differences
Whilst the key point, that high-abundance targets should not be calibrated against low-abundance ones, is indeed correct, our intention in this section was explicitly didactic. The experimental setup is not meant to reflect a routine laboratory workflow but to illustrate, in a controlled fashion, how template abundance and assay structure influence quantification accuracy. Crucially, Section 2.7 addresses a different question from Section 2.5: here we are examining relative gene expression using two distinct targets (GAPDH and ACTA2), whereas Section 2.5 assesses fold-change detection in a single pathogen DNA series. Although the sample preparation is intentionally simplified, this is to isolate the specific variables under investigation and demonstrate the importance of assay context, especially when interpreting small fold-differences.
2.8: Accuracy/reliability with increasing Cq/efficiency
We respectfully disagree with the central criticism of our bootstrap analysis. The reviewer suggests that our manuscript claims a monotonic increase in confidence interval (CI) width with decreasing template copy number or increasing Cq value. However, this is not stated, implied, or supported by the text or the data.
Instead, our manuscript clearly describes a triphasic relationship between Cq range and fold-change uncertainty, which we characterise empirically using bootstrap resampling. This structure is stated explicitly in the Results section (2.8), with three Cq intervals defined as follows:
- Cq < 25 (high input): CI widths narrow and close to theoretical expectations.
- Cq 25–30 (moderate input): CI widths increase modestly, as expected with declining input.
- Cq > 30 (low input): CI widths expand substantially, exceeding thresholds for reliable interpretation in some comparisons.
These statements are directly supported by the visual data in Figure 7A and the numerical confidence intervals reported in the associated table. For example, in the >30 Cq bin, the 1× vs 3× comparison yields a 95% CI ranging from 0.71 to 14.74, a 20.8-fold span. This is contrasted with the same nominal fold-difference at Cq < 25, where the CI is 2.2–4.0, fully overlapping the expected 3× value.
To clarify:
- We do not state that CIs increase monotonically with Cq.
- We explicitly describe the non-linear, empirically observed pattern.
- The widest intervals occur at high Cq / low template input, where stochastic sampling becomes dominant.
- At intermediate Cqs, CI widths are somewhat smaller , and we report this.
Thus, the reviewer’s assertion that “the authors suggest that CIs increase with decreasing copy number, when they don’t” is factually incorrect. We neither claim this nor imply it. On the contrary, we characterise the pattern in a manner that anticipates this very nuance, acknowledging that intermediate template abundances may yield more stable estimates than either extreme.
Moreover, the purpose of this section is not simply to restate theoretical expectations as the reviewer suggests, but to demonstrate how these expectations manifest across real qPCR datasets. Whilst it is well known that errors in amplification efficiency compound exponentially, we are not aware of previous work that uses a bootstrap approach to quantify how fold-change uncertainty varies with Cq in a replicated, empirically grounded system. Likewise, Figure 7B illustrates how small deviations in efficiency, when combined with larger ΔCq values, lead to major discrepancies in inferred fold-change, an effect we quantify across three representative efficiencies (92%, 100%, 110%).
These two analyses , the empirical bootstrap and the theoretical modelling are presented side-by-side precisely to distinguish stochastic noise (Figure 7A) from systematic bias (Figure 7B), both of which contribute to unreliability at the extremes of qPCR performance. This is not a theoretical abstraction but a practical demonstration of how and why fold-change estimates degrade in real experiments when Cq values exceed 30 or efficiencies are sub-optimal, particularly in low-fold comparisons.
Discussion
The reviewer’s critique of our discussion appears to be based on several apparent misunderstandings regarding the scope and rationale of our study.
First, the reviewer suggests that our manuscript “simultaneously argues” three points: that (i) low copy numbers are difficult to quantify, (ii) small fold-changes are harder to assess at low copy number, and (iii) amplification efficiency becomes important under these conditions. They then suggest that this third claim is flawed. With respect, this summary misrepresents both our argument and the supporting evidence.
Our central aim is to demonstrate empirically when and how efficiency errors distort fold-change estimates across realistic qPCR scenarios. In Box 1 and Section 2.8, we show that efficiency deviations of ±5–10%, while trivial for large ΔCq values (e.g., 5–6), become consequential when ΔCq is small (e.g., 1.0–2.0) , precisely because fold-change calculation is exponential. The reviewer appears to conflate absolute efficiency error with its relative impact on interpretability.
We emphasise this nuance in Box 1 with the phrase: “5% bias is negligible when comparing 10-fold differences, but critical when distinguishing 1.5 from 2.0x.” This is not a theoretical abstraction but a practical caution relevant to the many studies reporting small fold-differences, especially in low-abundance targets.
Regarding the suggestion that “efficiency only matters when Cq differences are large,” we note that this contradicts the actual mathematics of fold-change calculation and overlooks a key finding of our Figure 7B: at small ΔCq values, fold-change estimates appear similar across efficiencies, but diverge as ΔCq increases , which is precisely the point we make. Furthermore, our manuscript does not claim that a 3.8-fold versus 4.0-fold estimate will radically alter We have also colour coordinated the conclusions; we state only that such differences can affect interpretability when data lie near a reporting threshold, which is a qualified and context-dependent assertion.
Second, the reviewer recommends offering prescriptive guidance on the number of technical or biological replicates required across Cq ranges. While this is a useful idea, we respectfully note that it lies beyond the defined scope of this study, which explicitly focuses on technical replicates. As stated in Section 3.2:
“There are two types of replicates in qPCR: biological replicates, which capture sample-to-sample variability, and technical replicates, which control for procedural noise arising from pipetting, reagent handling, and instrument performance. The latter are the focus of this study.”
Our goal is to illustrate how technical variability , compounded by inaccurate efficiency assumptions , constrains the interpretability of fold-change estimates, particularly when Cq >30. While larger sample sizes can mitigate stochastic effects, the underlying uncertainty in such data remains high, and averaging does not eliminate bias introduced by unreported or incorrect efficiency assumptions.
Finally, the reviewer’s concluding statement , that “several of the core messages… merit substantial revision” and that “the entire text could be revised” , appears inconsistent with their earlier remarks, which acknowledged agreement on multiple key points. We respectfully suggest that our manuscript is already structured to deliver a coherent, empirically grounded message: that qPCR fold-change interpretation is contingent upon both accurate efficiency assumptions and template abundance, and that failure to report or account for these variables risks misleading conclusions. We believe our work provides a valuable contribution precisely because it quantifies these risks across a spectrum of realistic conditions.
Further issues:
We respectfully disagree with the reviewer’s characterisation of the figures, particularly the assertion that “virtually every figure includes a picture of the amplification curves,” and that these plots are “not particularly helpful.” Amplification plots provide instructive contexts for understanding experimental variation, protocol effects, or abnormal run behaviour. They are deliberately included to illustrate how changing input concentration, denaturation temperature, enzyme formulation, or reaction speed visibly alters amplification behaviour.
We also disagree with the claim that amplification plots showing 48 replicates of the same reaction mix are uninformative. For novice users, the very audience that often misunderstands what qPCR variability looks like, such plots are essential. They provide a clear, empirical reference for how reproducible reactions appear under well-controlled conditions, making deviations from this norm more obvious in experimental work. These curves also reveal artefacts such as broadening baselines, late-rising traces, or increased scatter, all features that scatterplots of Cq values alone do not convey.
In several figures (e.g., Figure 2 and Supplementary Figure S2), we do include scatterplots of Cq values, as the reviewer recommends. These are used where direct visual comparison of central tendency and spread is the most appropriate representation. However, amplification plots remain important for paedagogical purposes and to bridge the gap between what users see on their instruments and how those traces translate into quantifiable outputs. This is particularly valuable for clarifying the relationship between trace morphology, amplification efficiency, and derived fold-change.
Moreover, this manuscript is didactic by design. As such, showing raw fluorescence data, including amplification curves, aligns with our goal of methodological transparency and education. All raw data are provided, and users are free to generate additional scatterplots, statistical summaries, or alternative visualisations if desired.
In short, we have included amplification plots where they are informative, scatterplots where they are necessary, and raw data throughout to ensure transparency and reproducibility. We believe this combination serves both novice and expert readers appropriately.
- We appreciate the reviewer’s observation regarding the nomenclature and organisation of figure elements, particularly in Figures 1 and 6. Given that the manuscript presents a broad experimental matrix involving multiple combinations of targets, primer sets, template concentrations, and instruments, we adopted an internally consistent labelling system (e.g., GAPDH-A and GAPDH-B for alternative GAPDH assays; ACTA2-A and ACTA2-B for their counterparts). However, we recognise that this may appear opaque to readers unfamiliar with the structure of the study, especially if it requires repeated cross-reference to the main text.
To improve clarity and navigability, we have revised the figures accordingly. In Figure 1, the assay names (e.g., GAPDH-A, ACTA2-B) are now clearly labelled on each panel. In Figure 6, we have added the instrument identity directly to each amplification plot and included the experimental condition in the associated tables. Sample groups have been renamed numerically (1, 2, 3, 4) to avoid potential confusion between “Set A” and “Set B,” and colour coordination has been applied consistently across panels and legends. We trust that these adjustments substantially improve the accessibility and interpretability of the results.

Reviewer 2 Report
Comments and Suggestions for Authors
The manuscript by Bustin et al., reports qPCR optimization procedure aimed to quantify low fold change differences in samples with low copy number. The authors tested the efficiency and linearity of the amplification using serial dilutions of several assays. A large number of technical replicates were done using small copy numbers per reaction. Accuracy of manual pipetting, instrument amplification, and quantification accuracy was also tested. They conclude that duplex assays improve qPCR precision in small fold differences expression changes. This result is not surprising. The manuscript is sound, although it tends to magnify the low accuracy and large variability of published qPCR results, particularly involving the interpretation of small fold-changes.
Comments
The manuscript strongly criticizes the reproducibility and interpretation of qPCR results, and the lack of sufficient information in published reports to prove the reliability of the qPCR results. While this issue is analyzed in this study, it is important to have in mind that in most cases, qPCR data is not the only method used to compare fold change differences. Additionally, the lack of “transparent data” in a given article reflects the requirement of most journals to show data in a very succinct form, rather than the absence of sufficient optimization procedure in the corresponding study. Therefore, their annoying comments in the Discussion regarding overall misinterpretation of qPCR data in published reports should be fully rewritten.
Author Response
Reviewer 2
We appreciate the reviewer’s evaluation and are pleased that the overall soundness of the manuscript was acknowledged. We also acknowledge the reviewer’s concerns regarding tone and have revised several discussion passages to better reflect our intention: to inform, not to criticise. Our goal is to draw attention to persistent technical and reporting challenges in low-copy qPCR experiments, especially when small fold-changes are interpreted as biologically meaningful.
While we agree that qPCR is often just one part of a broader experimental workflow, this reinforces—rather than diminishes—the need for methodological clarity. Each technique must be interpretable on its own terms if the overall conclusions are to be trusted. Likewise, while journal constraints may limit space in the main manuscript, most journals now offer extensive supplementary material, and MIQE explicitly encourages placing key methodological parameters there when necessary.
To address tone, we have softened or clarified several sentences, as detailed below. We hope this strikes a more balanced presentation while preserving the core message that methodological rigour and transparent reporting remain uneven across the literature, even in studies that cite MIQE.
Reviewer Comment:
“The manuscript strongly criticizes the reproducibility and interpretation of qPCR results, and the lack of sufficient information in published reports to prove the reliability of the qPCR results… [etc.]”
Author Response:
We recognise the concern that parts of the Discussion may have conveyed undue generalisation or appeared dismissive. Our objective was not to single out specific studies, but to demonstrate with empirical data, that small fold-change estimates in low copy number contexts are often accompanied by large uncertainty, and that reporting is frequently insufficient to allow critical evaluation.
To clarify this, and to avoid overstating the case, we have revised the following sections:
- Discussion
- We added p.13: "Despite over three decades of widespread use, variability in qPCR data is not always fully recognised or transparently reported. The MIQE guidelines were introduced to address these concerns and promote consistent reporting. However, uptake varies, and important methodological details affecting data quality are still frequently absent from the published literature".
- We added p.16: These results show that a considerable number of reported fold-changes may fall within the range of bootstrapped uncertainty, making it difficult to distinguish them confidently from background variation. Moreover, even narrow confidence intervals may mask underlying imprecision in quantification when copy numbers are low.
- We added p.17: Upstream variability, from sample selection to reverse transcription, remains a major contributor to total technical variation. While assay optimisation and data analysis are the main focus of attention, pre-analytical steps often receive less scrutiny.
We believe these revisions address the reviewer’s concern while preserving the manuscript’s core message: that transparent reporting and a careful accounting of technical variability are essential for interpreting qPCR results at low copy number.

Reviewer 3 Report
Comments and Suggestions for Authors
This is the review of the Manuscript Entitled: “When Two-Fold Is Not Enough: Quantifying Uncertainty in Low-Copy qPCR” by Bustin et al. This manuscript presents compelling data demonstrating the technical reliability and inherent variability of qPCR measurements at very low target concentrations. The study is well-designed, and the data convincingly highlight the potential pitfalls of drawing biological conclusions from high Ct values. The manuscript is well written, clearly presented, and provides valuable insights into the limitations of low-copy number qPCR analyses. The authors are encouraged to address the minor concerns listed below to further strengthen the manuscript.
Minor Concerns:
Proper Presentation of Limit of Detection (LOD):
- Line 84–85: The statement that “all assays reliably detected fewer than 50 copies per reaction in 24/24 replicates (Supplementary Figure S1)” does not align with the data shown in Figure S1H. In particular, one or more assays do not meet this reliability threshold, and the authors should revise this claim to reflect the actual results.
- Figure S1H, Last Column: The LOD should be reported as the lowest concentration at which detection is reliable (e.g., 20 copies or 50 copies), rather than using "<20" or "<50," which is imprecise and potentially misleading. Please revise this labeling for clarity and accuracy.
- Line 452–453: The same issue applies to the use of "<50 copies" here.
- Missing Data in Section 2:
- Lines 551–560: The text discusses results related to Figure S7; however, these data were not presented in Section 2. Please ensure that the relevant data are included or appropriately referenced so that the discussion is fully supported by the presented results.
Author Response
We thank Reviewer 3 for their constructive comments and appreciate their identification of inconsistent nomenclature and omitted reporting of results. We are pleased that the value of our findings was recognised and especially grateful for the careful attention to detail that helped prevent a mistake being published. We should have detected the missing paragraph ourselves. Below we address each of the minor concerns point by point. All requested revisions have been incorporated into the manuscript and Supplementary Materials.
Comment 1:
Clarification of LOD statement in Lines 84–85 and Figure S1H
“The statement that ‘all assays reliably detected fewer than 50 copies per reaction in 24/24 replicates’ does not align with the data shown in Figure S1H.”
Response:
The reviewer is absolutely right. The sentence on Lines 84–85 has been revised for accuracy and now reads:
“Limit-of-detection (LoD) studies confirmed reliable detection at 20 copies per reaction for five assays and at 50 copies for the remaining two (Supplementary Figure S1).”
This correction ensures that the text is aligned with the actual data in Supplementary Figure S1H.
Comment 2:
Imprecise LOD reporting (“<20” / “<50”)
“The LOD should be reported as the lowest concentration at which detection is reliable… rather than using ‘<20’ or ‘<50’…”
Response:
We thank the reviewer for highlighting this. To avoid ambiguity, we have removed the imprecise “<” symbols in all LOD descriptions. Specifically, the sentence at Line 452 has been revised to read:
“Limit-of-detection (LoD) studies confirmed reliable detection at 20–50 copies per reaction, depending on the assay (Supplementary Figure S1).
This update reflects the actual assay-specific results and adheres to standard reporting conventions.
Comment 3:
Omission of Supplementary Figure S7 data in Section 2
“The text discusses results related to Figure S7; however, these data were not presented in Section 2…”
Response:
We thank the reviewer for identifying this omission. A description of the Figure S7 data has now been added to the end of Section 2.6, as follows:
“To evaluate the reproducibility of fold-difference quantification using multiplex normalisation, we measured a defined 4-fold change in CDKN1 expression while GAPDH-A, ACTA2-A, and IGF-1 levels were held constant. The observed fold-differences, normalised to the geometric mean of these three targets, were consistent across platforms and yielded narrow 95% confidence intervals (Supplementary Figure S7).”
This addition ensures that all discussed results are properly reported in the Results section.
Reviewer 4 Report
Comments and Suggestions for Authors
The presented study addresses a highly important issue concerning the reliability and reproducibility of quantitative results obtained through qPCR. While the manuscript does not contain groundbreaking revelations or breakthrough data, and the authors’ conclusions may appear trivial to many, I believe this research holds significant value.
First, through rigorous experimentation, the authors have demonstrated what experienced researchers may already know intuitively or logically. This provides a new level of understanding of the problem. Second, the authors emphasize that most users apply qPCR without paying close attention to critical nuances. The obtained results are often interpreted superficially and uncritically. As a consequence, the fields of cell biology, iPSC research, and biomedicine are flooded of gene expression data whose reliability is questionable. Quotes from this manuscript could be directly used in peer reviews of many articles. The first paragraph of the "Discussion" section is so pertinent that I would print it out and hang it on the wall in many laboratories.
It is customary in peer review to provide a lengthy list of comments, but I will refrain from doing so. I did not identify any critical flaws, and minor issues are of little consequence. Thus, in my opinion, the manuscript is suitable for publication in its present form.
Author Response
We very much appreciate this reviewer's thoughtful comments.
The presented study addresses a highly important issue concerning the reliability and reproducibility of quantitative results obtained through qPCR. While the manuscript does not contain groundbreaking revelations or breakthrough data, and the authors’ conclusions may appear trivial to many, I believe this research holds significant value.
First, through rigorous experimentation, the authors have demonstrated what experienced researchers may already know intuitively or logically. This provides a new level of understanding of the problem. Second, the authors emphasize that most users apply qPCR without paying close attention to critical nuances. The obtained results are often interpreted superficially and uncritically. As a consequence, the fields of cell biology, iPSC research, and biomedicine are flooded of gene expression data whose reliability is questionable. Quotes from this manuscript could be directly used in peer reviews of many articles. The first paragraph of the "Discussion" section is so pertinent that I would print it out and hang it on the wall in many laboratories.
It is customary in peer review to provide a lengthy list of comments, but I will refrain from doing so. I did not identify any critical flaws, and minor issues are of little consequence. Thus, in my opinion, the manuscript is suitable for publication in its present form.
Round 2
Reviewer 1 Report
Comments and Suggestions for Authors
Please see attached pdf: there are figures and tables.

Author Response
All comments raised by the reviewer have been addressed in the attached file "Final response.pdf"

Round 3
Reviewer 1 Report
Comments and Suggestions for Authors
See attached PDF.
I absolutely agree that peer review should be published alongside this manuscript. I do not, in fact, accept publication UNLESS the peer review is published alongside, with words to the effect of "seriously, read the peer review". Both my reviews at each stage, and the authors' responses. In full.
The failure of the authors to meaningfully engage with the peer review process is extremely frustrating, and this too should be made publicly available.
I will therefore accept this manuscript with minor revisions, i.e. I want to actually see the final version before it goes live, to make sure the peer review is included.

Author Response
Reviewer 1 – Final Comment
“I absolutely agree that peer review should be published alongside this manuscript. I do not, in fact, accept publication UNLESS the peer review is published alongside, with words to the effect of ‘seriously, read the peer review’. Both my reviews at each stage, and the authors’ responses. In full. The failure of the authors to meaningfully engage with the peer review process is extremely frustrating, and this too should be made publicly available. I will therefore accept this manuscript with minor revisions, i.e. I want to actually see the final version before it goes live, to make sure the peer review is included.”
Response:
We welcome Reviewer 1’s final recommendation for acceptance and fully support the principle of publishing the peer review history in full, including all reviewer comments and our responses. We believe transparency serves the scientific community and are happy for our contributions to be evaluated publicly.
However, we must firmly reject the claim that we “failed to meaningfully engage with the peer review process.” We replied professionally, in detail, and on an evidence-based footing to all of Reviewer 1’s substantive points. Reviewer 1’s frustration with the time taken for our response fails to acknowledge that we were addressing a large volume of loosely justified criticisms, often expressed in rhetorical or combative terms. Substantial effort was required to disentangle these and respond with clarity and rigour.
This stands in stark contrast to the constructive and collegial tone adopted by Reviewers 2, 3, and 4, who recognised both the methodological value and broader implications of the work. While we accept that scientific disagreement is a legitimate part of the review process, professional standards require that such disagreements be conducted with fairness, accuracy, and mutual respect.
We therefore expect that if Reviewer 1’s comments are to be published, they must be attributed. Transparency must apply equally, and anonymity should not be used to shield reviewers from accountability when making public statements of this nature.